# Immunoreactivity Analysis of MHC-I Epitopes Derived from the Nucleocapsid Protein of SARS-CoV-2 via Computation and Vaccination

**DOI:** 10.3390/vaccines12111214

**Published:** 2024-10-25

**Authors:** Dongbo Jiang, Zilu Ma, Junqi Zhang, Yubo Sun, Tianyuan Bai, Ruibo Liu, Yongkai Wang, Liang Guan, Shuaishuai Fu, Yuanjie Sun, Yuanzhe Li, Bingquan Zhou, Yulin Yang, Shuya Yang, Yuanhang Chang, Baozeng Sun, Kun Yang

**Affiliations:** 1Department of Immunology, The Key Laboratory of Bio-Hazard Damage and Prevention Medicine, Basic Medicine School, Air Force Medical University (In Former The Fourth Military Medical University), Xi’an 710032, China; superjames1991@foxmail.com (D.J.); mazilulu@fmmu.edu.cn (Z.M.); zjq000316@fmmu.edu.cn (J.Z.); sunyubo000103@fmmu.edu.cn (Y.S.); baitianyuan@fmmu.edu.cn (T.B.); liuruibo@fmmu.edu.cn (R.L.); wangyongkai@fmmu.edu.cn (Y.W.); guanliang@fmmu.edu.cn (L.G.); thisisfssxd@fmmu.edu.cn (S.F.); yuanjiesun@fmmu.edu.cn (Y.S.); 19721991974@fmmu.edu.cn (Y.L.); z17870236398@fmmu.edu.cn (B.Z.); yyl200404@fmmu.edu.cn (Y.Y.); yangsy@fmmu.edu.cn (S.Y.); cyh18295937463@fmmu.edu.cn (Y.C.); 2Yingtan Detachment, Jiangxi General Hospital, Chinese People’s Armed Police Force, Nanchang 330001, China; 3General Practice Medicine Base of Shanghai Changzheng Hospital, Shanghai 200041, China

**Keywords:** SARS-CoV-2, nucleocapsid protein (NP), MHC-I-restricted epitopes, CD8+ T cell response, immunogenicity

## Abstract

**Background**: Since 2019, the SARS-CoV-2 virus has been responsible for the global spread of respiratory illness. As of 1 September 2024, the cumulative number of infections worldwide exceeded 776 million. There are many structural proteins of the virus, among which the SARS-CoV-2 nucleocapsid (N) protein plays a pivotal role in the viral life cycle, participating in a multitude of essential activities following viral invasion. An important antiviral immune response is the major histocompatibility complex (MHC)-restricted differentiation cluster 8 (CD8+) T cell cytotoxicity. Therefore, understanding the immunogenicity of SARS-CoV-2 NP-specific MHC-I-restricted epitopes is highly important. **Methods**: MHC-I molecules from 11 human leukocyte antigen I (HLA-I) superfamilies with 98% population coverage and 6 mouse H2 alleles were selected. The affinity were screened by IEDB, NetMHCpan, SYFPEITHI, SMMPMBEC and Rankpep. Further immunogenicity and conservative analyses were performed using VaxiJen and BLASTp, respectively. EpiDock was used to simulate molecular docking. Cluster analysis was performed. Selective epitopes were validated by enzyme-linked immunospot (ELISpot) assay and flow cytometry in the mice with pVAX-NP_SARS-CoV-2_ immunization. Enzyme-Linked Immunosorbent Assay (ELISA) was used to detect whether the preferred epitope induced humoral immunity. **Results**: There were 64 dominant epitopes for the H-2 haplotype and 238 dominant epitopes for the HLA-I haplotype. Further analysis of immunogenicity and conservation yielded 8 preferred epitopes, and docking simulations were conducted with corresponding MHC-I alleles. The relationships between the NP peptides and MHC-I haplotypes were then determined via two-way hierarchical clustering. ELISA, ELISpot assay, and flow cytometry revealed that the preferred epitope stimulated both humoral and cellular immunity and enhanced cytokine secretion in mice. **Conclusions**: our study revealed the general patterns among multiple haplotypes within the humans and mice superfamily, providing a comprehensive assessment of the pan-MHC-I immunoreactivity of SARS-CoV-2 NP. Our findings would render prospects for the development and application of epitope-based immunotherapy in lasting viral epidemics.

## 1. Introduction

Coronavirus disease 2019 (COVID-19) is an infectious disease caused by severe acute respiratory syndrome coronavirus 2 (SARS-CoV-2). On 11 March 2020, the World Health Organization (WHO) declared this novel coronavirus a pandemic. Since December 2019, more than 776 million cases and 7.06 million deaths have been recorded globally, but the actual numbers are considered higher [1]. SARS-CoV-2 is distinguished by a spherical structure with a surface envelope comprising rod-shaped spikes. Among the 29 proteins encoded by these proteins, four are of particular importance: the spike protein (S), the membrane protein (M), the envelope protein (E), and the nucleocapsid protein (NP) [2].

The nucleocapsid protein (NP) is a structurally heterogeneous, 419 amino acid-long, multi-domain, RNA-binding protein that exists within the viral envelope. In virus-infected cells, it is the most prevalent protein. Research has shown that the SARS-CoV-2 NP acts as a viral suppressor of RNA silencing (VSR) through its double-stranded RNA binding activity to counteract host RNAi-mediated antiviral responses [3]. In addition, the NP triggers strong cellular and humoral responses in the host body after infection [4,5]. These characteristics make it a key target for developing vaccines.

The cellular immune response in the host after viral infection is mediated mainly by CD8+ T cells. The activation of the antiviral CD8+ T cell response requires antigen-presenting cells (APCs) to display viral antigens via MHC-I molecules. However, because of the inadequate availability of viral epitopes or the down-regulation of the host immune response by pathogenic viruses, the regulation of host CD8+ T cell presentation mediated by MHC-I is affected, and host immunity is sometimes still compromised [6]. Only high-affinity peptides are capable of triggering immune responses by binding tightly to MHC-I molecules on the APC surface [7]. Therefore, highly immunogenic epitopes are crucial for antiviral immunity [8]. Evolutionarily conserved antigenic epitopes are thought to help viruses survive, as they have the potential to confer broad protective immunity against these viruses, regardless of variant strains [9]. Recently, many vaccines and treatment plans have begun to design and implement virus epitopes at the core [10,11]. Vaccines designed on the basis of epitopes can improve safety and coverage by reducing adverse reactions, ultimately increasing vaccine efficacy [12,13].

Over the past few years, Verma and colleagues predicted three immunogenic NP peptides with high population coverage, and desirable docking with HLA-I molecules was identified [14]. Farhani and colleagues used conserved SARS-CoV-2 NP epitopes for vaccine production and obtained high population coverage that induced both cellular and humoral immunity. In addition, scientists have validated the functions of 120 SARS-CoV-2 CD8+ T cell epitopes and have used computers to predict high affinity and immunogenicity for peptide vaccines development [15]. Chao and colleagues confirmed NTASWFTAL with high immunogenicity which induced a strong CD8+ T cell response to SARS-CoV-2 [16]. Rigo and colleagues identified SPRWYFYYL as a conserved NP epitope [17], and Kumar and colleagues predicted five NP epitopes for vaccine construction [18]. The NTASWFTAL epitope has been ascertained to have HLA affinity and broad coverage [19,20]. In the limited context of immunogenetics, this field focuses mainly on predicting HLA-dominant epitopes. However, a comprehensive understanding of NP-specific pan-MHC-I biology is still lacking.

In the present study, we predicted the binding affinity and immunogenicity of these epitopes for the corresponding MHC-I molecules. The conservation of dominant epitopes has been confirmed in various isolates worldwide. Within wet-laboratory validation by ELISA, ELISpot assay, and flow cytometry, these sophisticated methodologies contributed to a better understanding of the SARS-CoV-2 NPs’ immunology while laying the foundation for the development of novel vaccines or immune modulators.

## 2. Materials and Methods

### 2.1. SARS-CoV-2 NP Sequence Retrieval

As input for the sequential in silico analyses, the nucleocapsid protein (N, accession number: NC_045512.2[28274.29533]) of SARS-CoV-2 (Wuhan strain) was obtained from the NCBI GenBank. To analyze where the amino acids vary among the SARS-CoV-2 strains and the differences in their affinity for the HLA molecules and the dominant 9-peptide, the protein sequences of reported isolated strains (84 nucleocapsid proteins in Appendix A) were obtained from NCBI GenBank.

### 2.2. SARS-CoV-2 NP Pan-MHC-I Epitope Prediction and Screening

Peptide candidates with high affinity were generated. We used well-adopted prediction algorithms to perform sequential oligo-peptide segmentation of the target NP sequence and calculate the affinity between MHC-I molecules. Eleven HLA-I genotypes and 6 mouse H-2 genotypes were included in the MHC-I molecules (Appendix A). To predict the binding affinity between each MHC molecule and the 9-mer peptide segment of SARS-CoV-2 NP, the following algorithms were applied: IEDB-recommended [21], SMMPMBEC [22], NetMHCpan4.1 [23], SYFPEITHI [24], and Rankpep [25,26] (For the comprehensive database versions, web server, and parameter settings, refer to Appendix A). Each algorithm displays the RANK values of an HLA molecule binding affinity to the NP-derived 9-mer peptides. We selected peptides that scored in the top 2% of more than two databases (when it comes to the MHC-I allele appearing in only four databases) and peptides that scored in the top 2% of more than three databases (when it comes to the MHC-I allele appearing in five databases).

### 2.3. Conservation Analysis

To determine the conservation of the predicted NP epitopes among different SARS-CoV-2 strains, we used BLASTP to conduct interspecific and intraspecific conservation analyses of all the predicted epitopes of SARS-CoV-2.

The evaluation standard for intraspecific protection is for SARS-related coronavirus (taxi: 694009), except for SARS-related acute respiratory syndrome coronavirus 2 (taxi: 2697049). The conservative evaluation standard between species was β-coronavirus (taxi: 694002), excluding SARS-related coronavirus (taxi: 694009). In the analysis, peptide sequences conserved between SARS-CoV-2 and humans (taxi: 9606) or mice (taxi: 10088) were also excluded, and the cutoff E value was <10^−5^. The predicted epitopes can be classified into four categories on the basis of their degree of conservation: intraspecific or interspecific conservation, both intraspecific and interspecific conservation, or neither intraspecific nor interspecific conservation.

### 2.4. Immunogenicity Analysis

A 9-mer peptide that exhibits high binding affinity with molecules such as MHC alone may not induce a sufficient immune response [27], as it requires high immunogenicity while possessing high immunoreactivity. The immunogenicity of the peptide segment itself is determined by its amino acid sequence. We calculated the immunogenicity of the 9-mer peptides in accordance with the recommended IEDB. A score > 0 was considered to indicate high immunogenicity [28].

### 2.5. Docking of Pan-MHC-I Molecules

Through the above methods, we obtained high-affinity, immunogenic, and well-conserved “preferred epitopes”. Next, we used computers to simulate the docking of these epitopes with human HLA-I and mouse H-2 molecules and selected the top ten docking modes with the lowest binding energy. In each docking mode, corresponding binding energies were obtained by the structural data from the RCSB PDB database for each MHC-I allele (Appendix A).

We used HPEPDOCK [29] to perform the docking of pan-MHC-I. Among the preferred epitopes and different binding modes of MHC molecules, the lowest binding energies were analyzed, and the average binding energies were calculated.

### 2.6. SARS-CoV-2 NP Peptides and Pan-MHC-I Clustering

Polymorphisms in MHC-I molecules, together with the diversity of the amino acid sequences of the epitopes, enable them to generate very diverse partnerships during the conjunction process to detect such relationships. We applied TBtools to perform bidirectional hierarchical clustering analysis on the affinity ranking data of the MHC superfamily and SARS-CoV-2-related 9-mer peptides [30]. The Z-score is calculated on the affinity ranking data before analysis. Euclidean distance hierarchical clustering was performed. The analysis revealed that 36 pan-MHC-I molecules interact with 411 SARS-CoV-2 NP epitopes, and a heatmap was generated to visualize this interaction.

### 2.7. Sequence Alignment of SARS-CoV-2 Variants

On the basis of the SARS-CoV-2 NP data, we obtained the results of ClusterX2.1 (Conway Institute UCD, Dublin, Ireland) for 84 variants. We utilized WebLogo to compare them [31]. The frequency of amino acid variation in different variants is represented by the height of the letters in the results. The results were analyzed, and the impacts of a specific amino acid mutation on all 9-mer peptide segments were drawn. The peptide segments with the highest mutation frequency were selected, and TBtools was used to create a heatmap of the delta differences in binding affinity between the SARS-CoV-2 NP and the variants of the corresponding HLA-I and 9-mer peptides. A positive RANK value indicates that the binding affinity of SARS-CoV-2 was greater than that of the variant, whereas a negative RANK value indicates that the variant has greater binding affinity than the original strain. Finally, we plotted a scatter plot of the binding affinity between the SARS-CoV-2 NP peptide segment T157H mutation and the original strain via Origin 2021 (OriginLab Corporation, Northampton, MA, USA). Finally, we analyzed mutated 9-mer peptides and determined whether they change in affinity due to the mutation.

### 2.8. Prediction of Peptide Toxicity and Sensitization

Peptides have proven to be among the most promising tools for the treatment and prevention of a wide range of diseases. However, their toxicity and sensitizing properties may lead to the development of a range of symptoms that can compromise the effectiveness of prevention and treatment. Therefore, studies of epitope toxicity and sensitization are important. We used network algorithms based on ToxinPred2 (https://webs.iiitd.edu.in/raghava/toxinpred2/index.html, accessed on 1 July 2024) and AlgPred 2.0 (https://webs.iiitd.edu.in/raghava/algpred2/, accessed on 1 July 2024) to test for toxicity and sensitization, respectively, of the screened “preferred epitopes”. Toxicity and lethality were set to standard thresholds (0.7 for negative toxicity; 0.4 for negative allergenicity). Machine learning models were used to output potential toxins and allergens. Epitopes were considered suspect if they exceeded a negative toxicity threshold of 0.7. Epitopes above a negative allergen threshold of 0.4 were considered to be suspected allergens and were tabulated for statistical analysis.

### 2.9. Application of Pan-MHC-I-Restricted SARS-CoV-2 NP Epitopes via a Literature Review

On the basis of previous research reports, we have summarized the use of SARS-CoV-2 NP-related epitopes. In both human and animal models, we have identified epitopes confirmed with cellular responses or antiviral protection. The dominant epitopes in this study were noted as a test of the innovation and feasibility of the experimental results.

### 2.10. Vaccine, Animal, and Immunization

The pVAX-NP_SARS-CoV-2_ vector was constructed in our laboratory. The gene encoding the SARS-CoV-2 NP was subjected to gene synthesis by TSINGKE (Tsingke Biotech Co., Ltd., Beijing, China) on the basis of a designed sequence. The BamHI cleavage site was introduced upstream, and the XhoI cleavage site was introduced downstream of the sequence. The SARS-CoV-2 *NP* gene was inserted into a pVAX1 vector to construct the pVAX-NP_SARS-CoV-2_ vector. The absence of mutations was verified by sequencing. After sequencing, it was confirmed that there was no mutation. The plasmid was purified via the Plasmid Maxi Kit (TIANGEN, Beijing, China) and stored at −20 °C until use. BALB/c and C57BL/6 mice at 8 weeks of age were obtained from the Laboratory Animal Centre of the Fourth Military Medical University. The mice were divided into four groups: the C57BL/6 experimental group, the BALB/c experimental group, and their respective blank control groups. Each experimental group contained six mice, and each blank control group contained three mice. At weeks 0, 3, and 6, pVAX-NP_SARS-CoV-2_ plasmids were subcutaneously injected at a dose of 50 μg per mouse. BALB/c and C57BL/6 mice injected with only PBS were used as blank controls. After each injection, we sacrificed two BALB/C and two C57BL/6 mice in each experimental group, as well as one BALB/C and one C57BL/6 mouse in each control group, and their spleen cells were collected for ELISpot experiments.

### 2.11. Peptides and ELISpot Assay

MHC-I restricted preferred epitopes of SARS-CoV-2 *NP* were artificially synthesized (ChinaPeptides, Shanghai, China). ELISpot experiments were used to identify CD8+ T cell immune responses produced by pVAX-NP_SARS-CoV-2_ inoculation. The ELISpot experiments used a 96-well plate with a PVDF membrane as the base, which was coated with specific anti-IL-2 monoclonal capture antibodies (diluted to 5 µg/mL (1:200) with sterile PBS). The mice were sacrificed, and the spleens were collected. The spleen was ground through a mesh screen and centrifuged. After red blood cell lysis for 10 min, RPMI 1640 containing 10% fetal bovine serum was added to the mixture, and the mixture was re-suspended to obtain mouse spleen cells. Under stimulation, T cells secrete IL-2 during the corresponding period. At this point, IL-2 is captured by antibodies encapsulated on the membrane. Then, the mouse spleen cells to be tested were added to the wells of the culture plate. They were stimulated with the treated SARS-CoV-2 NP peptides and cultured (diluted to 25 µg/mL (1:200) with sterile PBS). After the cells were washed away, the captured IL-2 bound to the biotin-labeled secondary antibody and then bound to biotin via streptavidin-HRP for chemical enzyme-linked colorimetry. Circular spots can form locally on the membrane, each corresponding to a mouse spleen cell that secretes IL-2. Next, 3-amino-9-ethylcarbazole (AEC; DAKEWEI, Shenzhen, China) was added to the HRP substrate, and the reaction was stopped by washing with water. The number of spots on the membrane was counted and then divided by the total number of cells added to the well to calculate the percentage of positive cells. The completed medium was used as the negative control in all four groups of splenocytes. As a positive control, Con A (10 µg/mL) was also used in all four groups of splenocytes. We used the S protein epitopes SARS-CoV-2 as irrelevant peptide control.

### 2.12. Enzyme-Linked Immunosorbent Assay (ELISA)

SARS-CoV-2 NP-specific antibodies in the mouse serum were detected via enzyme-linked immunosorbent assay (ELISA). The 96-well ELISA plates were incubated overnight at 4 °C with 10 μg/mL purified SARS-CoV-2 NP epitopes diluted in a coating buffer (1:1000 dilution, 10 µg/mL). After being blocked with PBS supplemented with 1% BSA for 2 h at 37 °C, the plates were washed with PBST (PBS supplemented with 0.05% Tween 20) four times. The immunized mouse sera were diluted (1:200, PBST with 0.1% BSA). Then, 100 μL of the diluted serum mixture was added to the wells and incubated for 1 h at 37°C. The plates were subsequently washed with PBST six times and incubated with HRP-conjugated goat anti-mouse IgG (CST, Hong Kong, China). After rinsing with PBST six times, the immune complex was developed by adding 100 μL of 3,3′,5,5′-tetramethylbenzudine (TMB) working solution (T0440, Sigma, Livonia, MI, USA). Finally, the reaction was ended with 50 μL of ELISA stop solution (C1058, Solarbio, Beijing, China), and the absorbance of the plates was read at 490 nm via a standard ELISA enzyme reader (Bio-Rad, Hercules, CA, USA). Overload of mouse serum was used as a positive control. Serum from a non-immunized mouse or PBS with 1% BSA was used as the reaction background or system negative control, respectively.

### 2.13. Flow Cytometry

The mouse splenocytes were harvested via the same method used for the ELISPOT. C57BL/6 mice elicited a stronger cellular response in the ELISPOT, so we continued to use the experimental and control C57BL/6 groups in flow cytometry studies. The mouse splenocytes were set into four groups: the medium control group of PBS-immunized mice, the NP peptide pool stimulation (the eight NP peptides were mixed and diluted with PBS, 10 µg/µL as a final concentration) group of PBS-immunized mice, and the two groups of mice that were immunized with pVAX-NP_SARS-CoV-2_ (with or without NP peptide pool stimulation). The mouse splenocytes were washed once with PBS and then twice with a flow washing solution containing 2% BSA. The washed lymphocytes were re-suspended and stained with a fluorescein-conjugated monoclonal antibody diluted in flow washing solution. The cells were incubated at 4 °C for half an hour in the dark. The cells were washed again with flow washing solution and placed in 300 µL of flow washing solution. Counting was performed with the NovoCyte flow cytometer and acquisition software NovoExpress 1.6.2 (ACEA Biosciences, Hangzhou, China). For data analysis, the different CD8+ T cell populations were gated in sequence. The first gate was FSC-A and SSC-H to detect single splenocyte distribution in the flow. For cytokine evaluation, a cell stimulation cocktail (Invitrogen, Carlsbad, CA, USA) was used. Both were incubated at 37 °C for 4 h. Cytofix/Cytoperm solution (BD) was used for fixation and permeabilization. The following antibodies (BioLegend, San Diego, CA, USA) were used for cytokine evaluation: CD3-FITC, CD8-Pacific blue, and IL-2/IFN-γ-APC.

### 2.14. Statistical Analysis

GraphPad Prism 9.0 software was used to analyze the data and visualize them. Statistical significance among different groups was evaluated via one-way ANOVA (* *p* < 0.05, ** *p* < 0.01, *** *p* < 0.001).

## 3. Results

### 3.1. Affinity Analysis of SARS-CoV-2 NP Epitopes for Mouse H-2 and Major HLA-I Haplotypes

A number of computational tools were used to perform the bioinformatics analyses (Figure 1). We obtained 238 HLA-I epitopes and 64 HLA-2 epitopes (Table 1 and Table 2). IEDB and NetMHCpan-4.1 provided the most coverage of the H-2 subtype. The dominant epitopes to the HLA-A3 allele claimed the greatest binding potential according to the results of the HLA-I alleles (48 peptides of the NP subtype, Table 1), and in the H-2 subtype, H-2 Db represented the most (19 peptides from the NP, Table 2).

We processed the RANK values with Z-score (data from NetMHCPan-4.1) and analyzed it with MHC molecules to construct a heatmap to show the regional affinity (as shown in Appendix A). The lower the RANK score, the greater the affinity of the epitope for the MHC-I molecule. Overall, the intensity of the epitopes showed a regional distribution.

### 3.2. Conservation Status of SARS-CoV-2 NP 9-mer Dominant Epitopes

To determine the conservation of the predicted epitopes between the protein sequences of different SARS-CoV-2 strains, we used BLASTP tools to conduct interspecific and intraspecific conservation analyses of all the predicted dominant epitopes of SARS-CoV-2. Table 3 shows the conservation of MHC-I-restricted dominant epitopes of SARS-CoV-2 NPs. Among them, there are two conserved peptide segments among multiple haplotypes. Eight dominant epitopes of SARS-CoV-2 NPs were interspecies−intraspecies+. None of them were interspecies+intraspecies−. According to the results, both interspecies and intraspecies-conserved multi-MHC-I reactive epitopes are more in human HLA-I than in mouse H-2 molecules.

### 3.3. Immunogenicity of SARS-CoV-2 NP 9-mer Peptides

The 9-mer epitopes that can induce a good immune response not only require high affinity but also need to be highly immunogenic. Among the 411 9-mer epitopes, 177 were immunogenic. Among these 177 epitopes, 47 were both immunogenic and of high affinity (Appendix A). After comparing their coverage of multi-MHC-I molecules reactiveness, we planned to synthesize high-affinity, immunogenic, and conserved 9-mer epitopes of SARS-CoV-2 NP and selected the final 8 preferred epitopes (Appendix A).

### 3.4. Interactions Between Pan-MHC-I Molecules and SARS-CoV-2 NP 9-mer Peptides via Hierarchical Clustering

The preferred epitopes cannot completely reflect the full picture of the SARS-CoV-2 NP during processing by pan-MHC-I haplotypes. To investigate the ability of different HLA molecules to bind to NP segments, we conducted bidirectional hierarchical clustering analysis on 411 SARS-CoV-2 NP 9-mer peptides (Figure 2).

Four clusters of 36 MHC-I subtypes were identified. These included 2 MHC-I exclusive clusters (HLA-I exclusive and H-2 exclusive) and 2 cross-reactive clusters (HLA major and H-2 major). Each MHC-I molecule was assigned to at least one cluster, except H-2 Dd, which independently stood out by itself (Appendix A).

### 3.5. Docking of Pan-MHC-I Molecules with Preferred Epitopes

After obtaining preferred epitopes with high affinity, immunogenicity, and conservation, we found that some of the preferred epitopes exhibited pan-MHC-I reactivity. These 9-mer epitopes can induce good immune responses across superfamilies and even species. Therefore, we used computers to simulate the docking of these peptides with different HLA-I molecules and mouse H-2 molecules.

Among the binding modes of each epitope and different MHC molecules, we analyzed the top 10 most favorable binding modes with the lowest binding energies (Appendix A). The average binding energies of these modes were lower for the epitopes LSPRWYFYY, SPRWYFYYL, and KHWPQIAQF than for the other epitopes, demonstrating favored docking performance and more stable chemical thermodynamic properties; that is, breaking the existing epitope structure requires more energy than for other epitopes. This finding may help infer that these 3 epitopes exhibit stronger affinity and immune reactivity. This characteristic is exhibited during the docking process with the human HLA-I subtype and the mouse H-2 allele. The scores of docking with human HLA-I subtypes in 4 of 8 epitopes were lower than those docking with mouse H-2 alleles, indicating favorable docking performance in the human body (upper Figure 3 and Appendix A). The other 4 epitopes, such as LSPRWYFYY, tended to bind to the mouse H-2 allele, but the differences in scores were relatively small, referred to in lower Figure 3 and Appendix A.

### 3.6. Multiple Sequence Alignment with 84 SARS-CoV-2 Mutant Strains

On the basis of 8 high-affinity segments, we determined the frequency of variation among SARS-CoV-2 NPs and the other 84 variants. Figure 4A shows the four mutations (T157H, D217E, A218T, and I336T). The frequencies were 85.89% for the T157L mutation, 81.18% for D217E, 81.18% for A218T, and 12.94% for I336T (Figure 4B).

### 3.7. Differences in the Immunoreaction Among SARS-CoV-2 and Its Variants

Mutations among SARS-CoV-2 and its 84 variants resulted in differences in binding affinity (accession no. in Appendix A). Mutant strains were selected from 4 virus species in the genus β-coronavirus: severe acute respiratory syndrome coronavirus 2 (SARS-CoV-2), bat coronavirus, *Sarbecovirus* sp., and severe acute respiratory syndrome-related coronavirus. Their numbers are 8, 5, 70, and 1, respectively.

The segment aa148–166 showed the most significant difference and was selected for further analysis. The most significant mutation, T157H, accounted for 85.89% of the mutation frequency. Figure 4C shows a heatmap of the differences in the binding affinities of all affected peptide segments before and after mutation of epitope LTYTGAIKL, and Figure 4D shows a scatter plot of the differences in the binding affinities of aa148--aa166. In the binding of YTGAIKLDD to HLA-A1 and HLA-B58, the original SARS-CoV-2 NP strain showed stronger affinity, whereas in the binding of YHGAIKLDD to HLA-24, the variants showed enhanced affinity to the original one. For SGTWLTYTG, the binding affinity with the variants was greater than that of the original strain, whereas the binding affinities of PSGTWLTYH, GTWLTYHGA, and TYHGAlKLD for the original strain remained greater than those for the variants. There was a single mutation in 4 epitopes that resulted in 3 changes in the affinity of HLA-I binding, in which 3 of them were strengthened in variants, but the remaining mutation had the opposite effect (Table 4).

### 3.8. Toxicity and Sensitization Analysis of SARS-CoV-2 NP Epitopes

Toxicity predictions were performed via two models of the website (ML+Hybrid). Sensitivities were predicted via the (MERCI+BLAST) model for websites. The toxicity and sensitization of 8 NP 9-mer epitopes screened via a bioinformatics network algorithm were analyzed separately. Under the default threshold (≤0.7 for negative toxicity and ≤0.4 for negative sensitization), LALLLLDRL was identified by the algorithm as being at risk of causing toxicity among the dominant epitopes screened previously. NNAAIVLQL and DAALALL are not only toxic but also allergenic (Table 5). Three peptides have low toxicity effects, and the sensitivity effect of DAALALLLL is also very low. Only NNAAIVLQL has greater sensitization. Thus, despite the risks associated with its application, it is still an epitope with great potential.

### 3.9. ELISpot Validation of the SARS-CoV-2 NP Epitopes

For experimental verification, eight H-2-restricted immunogenic dominant epitopes were synthesized. The mice were stimulated with single or pooled preferred epitopes. After 24 h incubation, the secretion of IL-2 from the spleen cells was observed. As showed in Figure 5, C57BL/6 mice presented stronger immune responses than BALB/c mice did after NP-derived epitope stimulation. The epitope NNAAIVLQL induced the strongest cytokine secretion in both types of mice, followed by NTASWFTAL, LSPRWYFYY, and SPRWYFYYL.

### 3.10. SARS-CoV-2 NP Epitopes Enhanced the Secretion of IFN-γ and IL-2

To elucidate whether the preferred epitopes induced an NP-specific CD8+ T cell immune response, we used flow cytometry to detect population expansion and cytokine secretion from CD8+ T cells. Figure 6A shows the gating strategy. Figure 6B shows the number of CD8+ T cells sub-population. Figure 6C shows CD8+ T cells IFN-γ, where the pVAX-NP_SARS-CoV-2_-immunized (NP peptide pool stimulation) group secreted the most IFN-γ (41.35%). The control group with the culture medium alone secreted the lowest amount of IFN-γ (21.83%), which reflected the background response levels. Figure 6D shows the responses of CD8+ T cells’ IL-2 secretion in the spleens of immunized mice. Among them, the group immunized with pVAX-NP_SARS-CoV-2_ (with NP peptide pool stimulation) produced the most IL-2 (11.06%), which was significantly greater than that of the blank control group (6.16%).

The results showed that mice injected with pVAX-NP_SARS-CoV-2_ not only produced more CD8+ T cells but also induced them to secrete more cytokines in an NP-derived epitopes’ specific manner. Compared with the PBS group, responses were significantly strengthened by the pVAX-NP_SARS-CoV-2_ injection. The group stimulated with the pooled NP peptides also produced more CD8+ T cells and more cytokines.

### 3.11. The Preferred Epitopes Induced Humoral Immune Responses

The corresponding antibodies were detected for all 8 preferred epitopes (Figure 7). Differences in the absorbance values for the 8 preferred epitopes were present but not statistically significant. The trends in the differences in absorbance values for C57BL/6 and BALB/c mice were similar, suggesting that activation of the humoral immune response by NP was not affected by H-2 subtypes. This phenomenon was not affected by the type of mouse, but stronger light absorbance values were detected in BALB/c mice than in control mice, and previous studies have shown that BALB/c mice do trigger stronger humoral immune responses, which was confirmed by our experiments. Among them, epitopes 216 and 219 were more immunogenic in activating the B-cell immune response. This may be related to the fact that these 2 epitopes share the same B-cell epitope, which requires further investigation.

## 4. Discussion

In previous studies, laboratories in various regions screened good SARS-CoV-2 NP epitopes from multiple perspectives, such as immunogenicity, affinity, and conservation [14,32,33]. In the limited context of immunogenetics, this field focuses mainly on HLA-dominant epitope prediction research or selects longer peptides when selecting peptide segments without accurately locating them to the 9 peptides. There is still a lack of comprehensive understanding of the high-affinity docking of specific peptide segments and HLA molecular or cross-racial binding similarity. In our study, we explored all 9-mer epitopes of SARS-CoV-2 NPs and identified eight high-affinity, immunogenic, and conserved CD8 epitopes, which we refer to as pan-MHC preferred epitopes. We subsequently simulated the docking of these pan-MHC preferred epitopes with MHC-I and H-2, obtaining more precise binding positions between MHC molecules and peptides while verifying the trend of the immune response across species. Afterwards, we investigated the interactions between all NP peptide segments of SARS-CoV-2 and MHC molecules through biphasic hierarchical clustering analysis, revealing the similarities in interactions between different MHC molecules, superfamilies, and even across species. Additionally, when SARS-CoV-2 NPs and their 84 variants were compared, a total of 26 amino acid site variations involving high-affinity and immunogenic epitope peptides were observed. Among the eight pan-MHC- preferred epitopes, only four amino acid variation sites were involved, and the overall binding potential of the MHC-I superfamily was minimally affected.

Since the emergence of SARS-CoV-2, neutralizing antibodies that specifically bind to SARS-CoV-2 have been the focus of scientists’ research [34]. However, the specific CD8+ T cell response, a key component of the human antiviral response, is an indispensable pathway for clearing viruses within host cells [35,36] and is also indispensable for the development of vaccines [37,38]. Several regions of the SARS-CoV-2 NP can be recognized by CD8+ T cells. They induce a strong cellular immune response [39]. High levels of IL-2 were detected in the epitopes obtained through our multidimensional exploration in the ELISpot experiment, revealing the preferred epitope response of COVID-19 NP-specific CD8+ T cells. In addition, we further verified via flow cytometry that the preferred epitope can stimulate the production of stronger CD8+ T cell responses and a certain amount of IL-2 and IFN-γ. These findings suggest that this approach could be used to develop protective CTL epitope vaccines.

Only antigens that possess both immunoreactivity and immunogenicity can trigger an immune response. Therefore, immunogenicity analysis has become an indispensable part of epitope research. The analysis of the immunogenicity of epitopes is essential and is gradually being applied in various viral studies on epitopes and vaccines [40,41,42]. In our study, we first used the IEDB tool to perform immunogenicity analysis on high-affinity epitopes. A total of 41 high affinity and immunogenic epitopes were screened. Eight high-affinity, high-immunogenicity, and high-conservation pan-MHC preferred epitope peptides were obtained. Among them, NTASWFTAL has been validated by Chao and colleagues as a highly immunogenic peptide segment [16], and our results support this conclusion.

Understanding the structure of epitope complexes can help us understand the molecular mechanisms of related biological processes, which is highly important for the development of peptide drugs. Therefore, we applied HPEPDOCK for molecular docking simulations [29]. We docked eight preferred epitopes to human HLA-I and mouse H-2 molecules. These epitopes can dock well with both human and mouse MHC molecules, and seven of them have a lower binding energy in docking with human HLA-I class molecules than H-2 does, demonstrating better docking performance. As the bidirectional hierarchical clustering heatmap shows, the affinity of mouse H-2 Db and H-2 Kb molecules in the docking simulation is similar to that of human HLA-A24 and HLA-A2 when binding to epitopes, whereas mouse H-2 Ld has a similar affinity for HLA-B7. This is highly important for further vaccine development. The average binding energy of the nine peptide epitopes, LSPRWYFYY, SPRWYFYL, and KHWPQIAQF, during the docking process was lower than that of the other epitopes, demonstrating good docking performance and more stable chemical thermodynamic properties. That is, breaking the existing epitope structure requires more energy than breaking other epitopes. This finding may help infer that these three epitope peptides exhibit stronger affinity and immune reactivity than other epitopes do. In subsequent ELISpot experiments, these three epitope peptides also induced good IL-2 responses similar to the predicted results and verified their accuracy. Peptides such as KTFPPTEPK and NTASWFTAL are considered to have high affinity or immunogenicity [16,19], which is also consistent with our predicted experimental results.

Human genetic factors are associated with susceptibility to and severity of SARS-CoV-2-induced disease [43,44]. One HLA-I allele (B*35:01) was associated with a shorter duration of COVID-19 in Caucasian patients infected with SARS-CoV-2 [45]. Ecuadorian COVID-19 patients who carry the HLA-A*24:02 allele may be protected from more severe forms of COVID-19 [46]. Given this potential association, our study of pan-MHC-I molecules included 30 members of the human HLA-I superfamily and 6 mouse H-2 alleles, covering HLA-A*24:02 and HLA-B*35:01, which were mentioned earlier, and achieved wide geographic and community coverage. Moreover, we combined five prediction algorithms to improve the accuracy of the prediction results when predicting advantageous epitopes. In our study, epitope peptides with cross-reactivity exhibited effective conditions and similar postures when docking with MHC-I molecules in both humans and mice, which seem to exceed the limitations of the MHC-I. In previous studies, we reported that the dominant epitope distribution of H-2d is similar to that of the human HLA-II superfamilies [47]. Bidirectional hierarchical clustering analysis revealed that human HLA-I molecules are clustered with patterns when they bind to the same epitope, which also occurs when it comes to both HLA-I and H-2 molecules at the identical epitope. For example, H-2 Db and Kb have very similar binding patterns to human HLA-A2 and HLA-A24. The scope of application is limited, which guarantees that specific epitope should be further evaluated. The fraction of the SARS-CoV-2 NP epitopes that bound to the MHC-I molecules of humans and mice was analyzed. We conclude that C57BL/6 mice may serve as a more suitable alternative to SARS-CoV-2 experimental models in the absence of humanized HLA-I transgenic mice.

SARS-CoV-2 originates from bats and other mammals [48]. When they are in close contact with the exchange virus, the coronavirus undergoes recombination, leading to diversification and the evolution of highly effective strains that are susceptible to human infection [49]. It continues to branch and evolve among humans worldwide [50]. The evolution of viral variants may negatively affect the immune system established by existing COVID-19 vaccines. These mutant strains are more contagious and more susceptible to reinfection [51]. Some mutant strains can also affect the efficacy of vaccines [52]. There is a low level of homology between SARS-CoV-2 variants worldwide [53]. Therefore, it is necessary to study the variation in viral antigens. Among the SARS-CoV-2 virus surface proteins, the main target for vaccine development is the S protein [54], but its epitopes are likely to be lost. The NP is more conserved than the S protein and thus more likely to provide sustained protection against COVID-19 when faced with new variants [55]. Our research revealed little difference in the binding affinity between original SARS-CoV-2 NP-preferred epitopes and their 84 variants, and only three out of the eight varied. This could be due to the high conservatism of the NP itself. Although N proteins are highly conserved, progressive escape mutations in the N sequence affect their conservation properties. This fact would be a limitation for the development of cross-reactive vaccines based on N proteins [56].

On the basis of the highly conserved SARS-CoV-2 NP epitopes that are shared by human coronaviruses, cross-protective cellular immunity against SARS-CoV-2 and its variants can be established [57]. This means that the extent of protective immunity can be increased by conserved epitopes of the immune response of interspecies viruses. Therefore, different intraspecific and interspecific conserved epitopes can expand the range of protection and provide broad protective immunity for both new and old SARS-CoV-2. In our study, epitopes were classified into four categories on the basis of whether they were conserved within or between species. Among the eight preferred peptide segments, two conserved/interspecies epitopes, namely LSPRWYFYY and SPRWYFYYL, are considered highly valuable for the development and research of epitope vaccines. Interestingly, during molecular docking simulations, these two peptide segments also exhibited better affinity and immunoreactivity than the other peptides did. The preferential T cell cross-reactivity of LSPRWYFYY, SPRWYFYYT, and their homologs from seasonal coronaviruses indicates persistent protective immunity [58]. SPRWYFYYL has also been identified as a conserved peptide by Mauricio Menegatti Rigo et al. [17]. We speculate that these peptides may be further applied in subsequent vaccine development and immunological experiments.

Epitope-based vaccine design is an effective method that deserves to be noticed. It can induce a strong and specific immune response against dominant epitopes and prevent the side effects of intact antigens on the human body. For example, tandem epitope vaccines have been designed for both NP and membrane proteins [59], or based on spike proteins, nucleocapsid proteins, and membrane proteins [60,61]. These vaccines contain nine 9-mer peptides, including our preferred epitopes of LSPRWYFYY, NTASWFTAL, and SPRWYFYL. Another multi-epitope vaccine based on the SARS-CoV-2 NP design applied a total of 14 CTL epitopes, of which 9 were predicted to have high affinity and immunogenicity [18], further confirming the accuracy of the prediction results.

However, the limitations of prediction methods cannot be ignored, such as in hierarchical clustering analysis, where our calculations do not consider the approximation and interconnectivity of the data. Despite its shortcomings, we still propose a method for screening advantageous preferred epitopes on the integration of multiple algorithms and databases, which not only reduces the accidental errors caused by a single database but also reduces the waste of human resources and errors caused by repeated experiments. The comparison between the predicted results and subsequent experiments further increases the accuracy of the results. Our study also provides insight into the extensive protection between different variants, providing guidance for the development of highly protected epitope vaccines in the future. Moreover, the study of different viral epitopes and vaccine development processes is also discussed in the context of viral mutations. The final product, which is proposed to be suitable for vaccine development and other uses, would only be obtained in subsequent in vitro/in situ studies.

## 5. Conclusions

Given that the NP is highly valuable in SARS-CoV-2 viral immunology and clinical application, we performed bioinformatics analysis via various databases to screen 9-mer peptides with high affinity and high immunogenicity. Inter- and intraspecies conservation analyses subsequently revealed eight preferred epitopes. Hierarchical clustering analysis revealed similarities in interactions among different MHC-I molecules, superfamilies, and even species. Molecular docking allows visualization of the docking of MHC molecules to epitopes. In addition to experimental validation and cross-lineage responsiveness exploration, the antigenic properties revealed the antiviral applicability of the NP and its epitopes in SARS-CoV-2 prevention and control.

## Figures and Tables

**Figure 1 vaccines-12-01214-f001:**
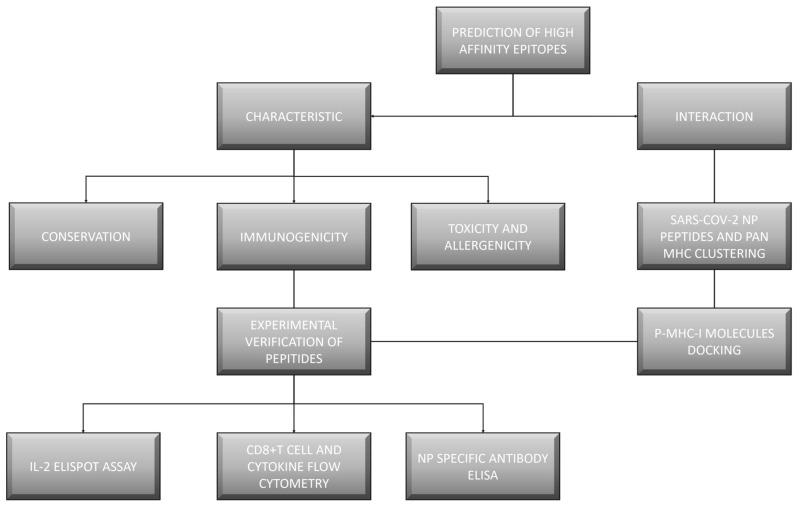
Flowchart of the experiment.

**Figure 2 vaccines-12-01214-f002:**
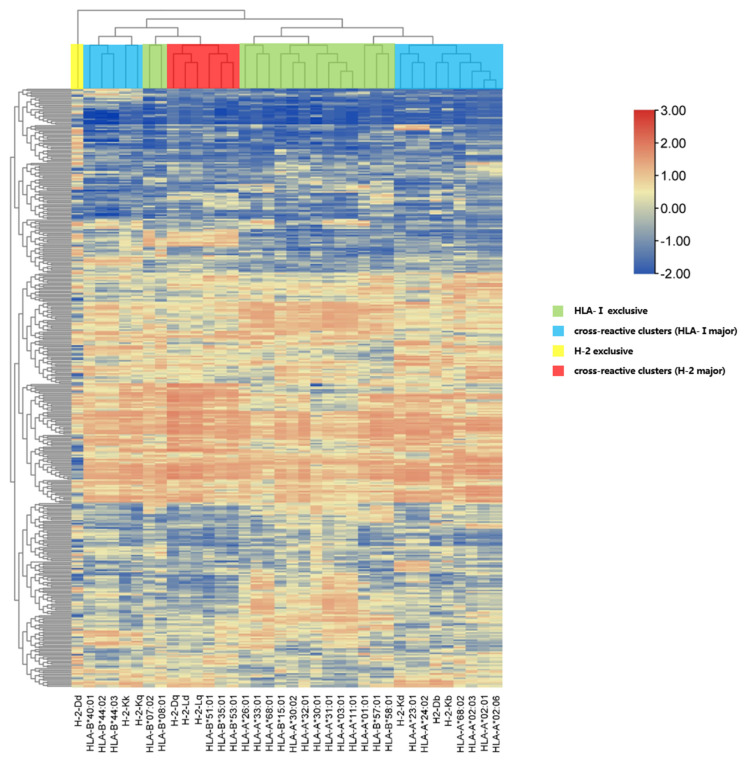
Hierarchical clustering analysis of all the predicted 9-mer epitopes in SARS-CoV-2 NPs. The yellow area represents the H-2 exclusive zone, with one H-2 subtype. The red area is the cross region between H-2 and HLA-I molecules, which is mainly composed of H-2 molecules, including three subtypes of HLA-B7 and H-2. The blue area is the cross region between HLA-I and H-2 molecules, which are mainly composed of HLA-I class molecules. The blue area on the left mainly includes two subtypes, HLA-B44 and H-2. The blue area on the right mainly includes three subtypes: HLA-A2, HLA-A24, and H-2. The green area is exclusive to HLA-I and only includes HLA-I subtypes. Similarly, neighboring MHC class I molecules have similar extraction capacities. In the heatmap, strong binding affinity is represented in red, and weak binding affinity is represented in blue.

**Figure 3 vaccines-12-01214-f003:**
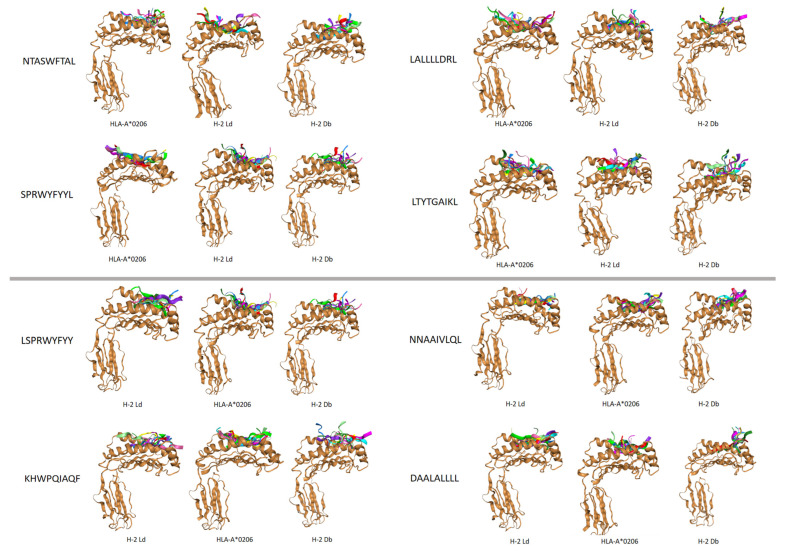
The top ten candidates with the highest scores for MHC docking are shown here. The top four tended to bind to human HLA-I molecules, and the bottom four tended to bind to mouse H-2 molecules. Different colors represent different modes of epitope binding.

**Figure 4 vaccines-12-01214-f004:**
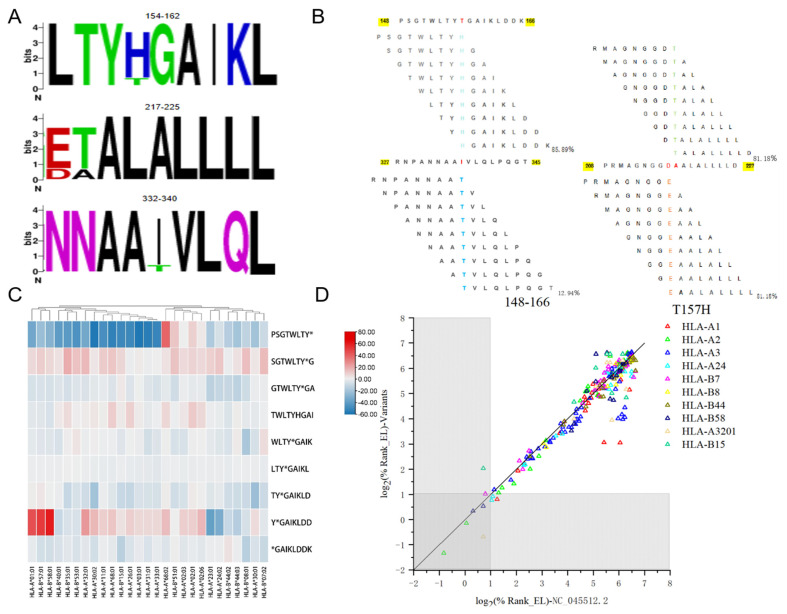
Figure (**A**) Comparison between the 9-peptide segment of the SARS-CoV-2 NP mutant and the original peptide segment. Figure (**B**) shows the high-frequency mutation sites and frequencies of 84 variants. Figure (**C**) shows a heatmap of the differences in binding affinity of all affected peptide segments before and after mutation of the high-frequency mutant peptide LTYTGAIKL. (Blue indicates a higher binding affinity of the original SARS-CoV-2 strain than the mutant, and red indicates a higher binding affinity than the original SARS-CoV-2 NP strain.) Figure (**D**) shows a scatter plot of the differences in the binding affinities of aa148–aa166. (Each nonapeptide episode is represented by a point in the scatter plot, and the gray areas indicate that this episode’s affinity ranking is in the top 2%.) The closer the line in the image is to y = x, the smaller the variation in binding affinity before and after mutation.

**Figure 5 vaccines-12-01214-f005:**
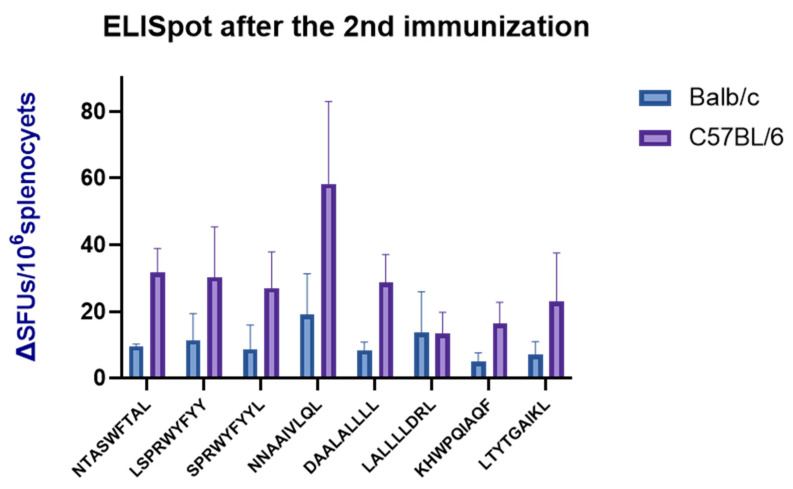
Validation of the dominant immunogenic epitopes by ELISpot experiments. The values used in the analysis results are the spots number of the pore minus the spots number of the corresponding negative control. Mouse spleen cells are stimulated to mount cellular immune responses via the 8 dominant epitopes. Blue represents IL-2 secreted by BALB/c mice, and purple represents that secreted by C57BL/6 mice.

**Figure 6 vaccines-12-01214-f006:**
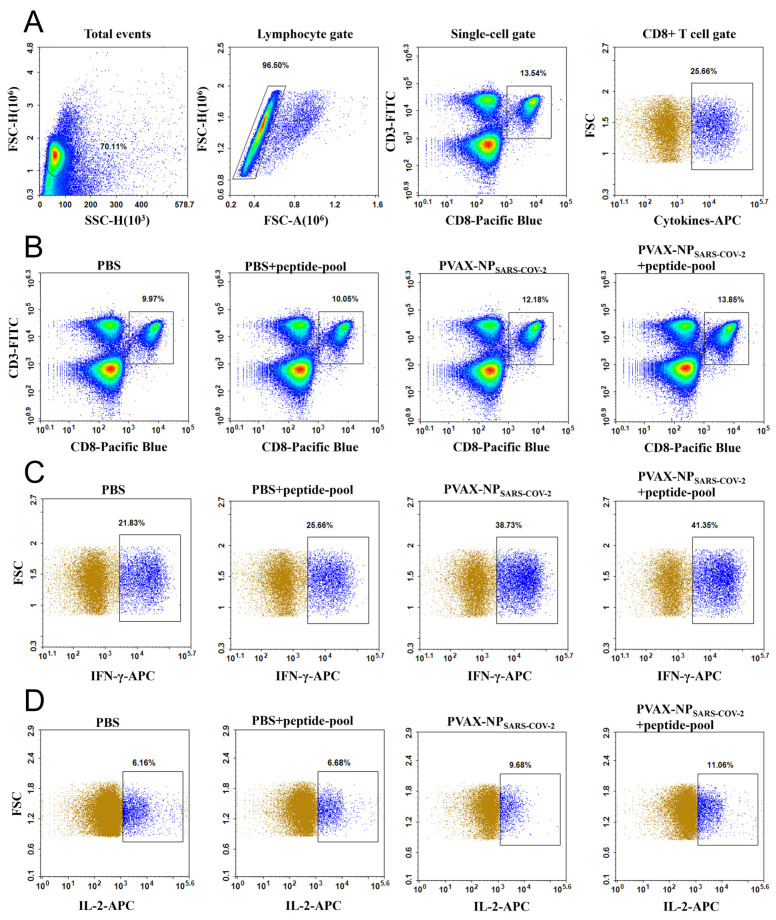
The expression of the CD8+ T cell cytokines IL-2 and IFN-γ was examined via flow cytometry. (**A**) Gating strategy diagram for CD8+ T cells sub-population and cytokine detection (the Cytokine-APC refers to a testimony of IFN-γ gating). (**B**) The number of CD8+ T cells in mice. (**C**) Enhanced CD8+ T cells’ IFN-γ secretion in mice that had been immunized or/with pooled peptide stimulation. (**D**) Enhanced CD8+ T cells IL-2 secretion in mice that had been immunized or/with pooled peptide stimulation.

**Figure 7 vaccines-12-01214-f007:**
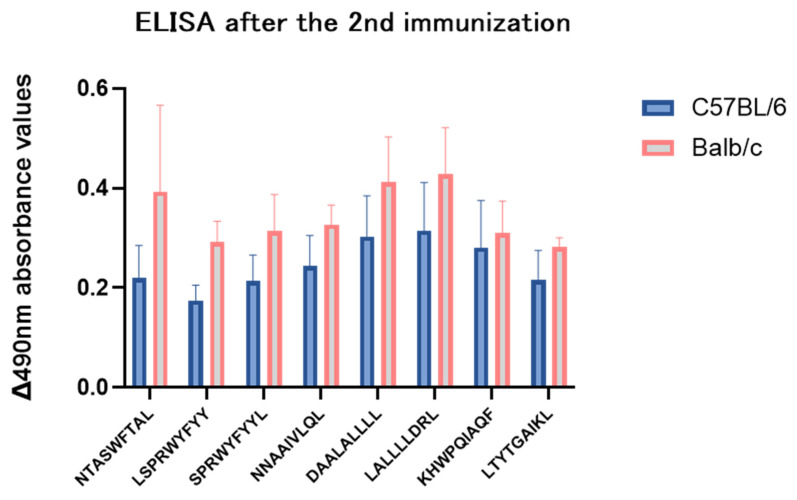
The preferred epitopes can be significantly recognized and bound to specific SARS-CoV-2 NP antibodies.

**Table 1 vaccines-12-01214-t001:** Numbers of HLA-1-dominant epitopes in SARS-CoV-2 NPs.

MHC-IHaplotypes	Prediction Tools	NP Epitopes	NP (Short-Listed)
HLA-A1	IEDB	38	25
	NetMHCpan	25	
	Rankpep	9	
	SMMPMBEC	11	
	SYFPEITHI	0	
HLA-A2	IEDB	25	21
	NetMHCpan	23	
	Rankpep	27	
	SMMPMBEC	6	
	SYFPEITHI	8	
HLA-A3	IEDB	57	48
	NetMHCpan	49	
	Rankpep	28	
	SMMPMBEC	25	
	SYFPEITHI	16	
HLA-A24	IEDB	18	16
	NetMHCpan	14	
	Rankpep	9	
	SMMPMBEC	7	
	SYFPEITHI	8	
HLA-3201	IEDB	14	19
	NetMHCpan	19	
	Rankpep	0	
	SMMPMBEC	5	
	SYFPEITHI	0	
HLA-B7	IEDB	42	22
	NetMHCpan	33	
	Rankpep	15	
	SMMPMBEC	19	
	SYFPEITHI	32	
HLA-B8	IEDB	18	10
	NetMHCpan	13	
	Rankpep	0	
	SMMPMBEC	5	
	SYFPEITHI	0	
HLA-B15	IEDB	23	17
	NetMHCpan	15	
	Rankpep	0	
	SMMPMBEC	4	
	SYFPEITHI	8	
HLA-B44	IEDB	23	15
	NetMHCpan	15	
	Rankpep	9	
	SMMPMBEC	2	
	SYFPEITHI	16	
HLA-B58	IEDB	17	11
	NetMHCpan	11	
	Rankpep	15	
	SMMPMBEC	6	
	SYFPEITHI	8	
HLA-B46	IEDB	21	15
	NetMHCpan	15	
	Rankpep	0	
	SMMPMBEC	4	
	SYFPEITHI	0	
HLA-B62	IEDB	0	
	NetMHCpan	23	
	Rankpep	0	
	SMMPMBEC	0	
	SYFPEITHI	0	
HLA-C0401	IEDB	22	19
	NetMHCpan	16	
	Rankpep	0	
	SMMPMBEC	5	
	SYFPEITHI	0	

NP epitopes are those that are in the top 2% of the results of each algorithm; NPs (shortlisted) are those that appeared in at least three predicting algorithms.

**Table 2 vaccines-12-01214-t002:** Numbers of murine MHC-I-dominant epitopes of SARS-CoV-2 NPs.

MHC-IHaplotypes	Prediction Tools	NP Epitopes	NP (Short-Listed)
H-2 Db	IEDB	15	9
	NetMHCpan	10	
	Rankpep	9	
	SMMPMBEC	5	
	SYFPEITHI	8	
H-2 Dd	IEDB	31	19
	NetMHCpan	15	
	Rankpep	9	
	SMMPMBEC	3	
	SYFPEITHI	0	
H-2 Kb	IEDB	19	13
	NetMHCpan	11	
	Rankpep	9	
	SMMPMBEC	6	
	SYFPEITHI	0	
H-2 Kd	IEDB	20	7
	NetMHCpan	13	
	Rankpep	9	
	SMMPMBEC	4	
	SYFPEITHI	8	
H-2 Kk	IEDB	11	9
	NetMHCpan	7	
	Rankpep	9	
	SMMPMBEC	0	
	SYFPEITHI	8	
H-2 Ld	IEDB	20	7
	NetMHCpan	16	
	Rankpep	9	
	SMMPMBEC	6	
	SYFPEITHI	8	

NP epitopes are those that are in the top 2% of the results of each algorithm; NP (shortlisted) are those that appeared in at least three predicting algorithms.

**Table 3 vaccines-12-01214-t003:** Conservation of MHC-I-restricted dominant epitopes of SARS-CoV-2 NPs.

MHC-I Haplotypes	Interspecies− Intraspecies−	Interspecies− Intraspecies+	Interspecies+ Intraspecies−	Interspecies+ Intraspecies+
H-2 Db	9	0	0	0
H-2 Dd	14	4	0	1
H-2 Kb	9	2	0	2
H-2 Kd	7	0	0	0
H-2 Kk	8	1	0	0
H-2 Ld	5	1	0	1
HLA-A1	21	3	0	1
HLA-A2	20	0	0	0
HLA-A3	45	2	0	0
HLA-A24	13	3	0	0
HLA-3201	10	4	0	0
HLA-B7	20	1	0	1
HLA-B8	4	0	0	0
HLA-B15	8	0	0	0
HLA-B44	11	4	0	0
HLA-B58	7	3	0	1
HLA-B4601	13	1	0	1
HLA-C0401	17	1	0	0

**Table 4 vaccines-12-01214-t004:** Changes in epitopes and their associated molecules.

Amino Acid	NC_045512.2	Variants	Dominant in	Dominant in	HLA-Ⅰ
Number			Variants	NC_045512.2	Genotype
			YES	NO	HLA-A2402
151–159	TWLTYTGAI	TWLTYHGAI	YES	NO	HLA-A2301
			YES	NO	HLA-A2601
153–161	LTYTGAIKL	LTYHGAIKL	NO	YES	HLA-B5101

**Table 5 vaccines-12-01214-t005:** Comprehensive assessment of the toxicity and allergenicity of the dominant epitopes.

Pepitides	MERCI Score	BLAST Score	Prediction	ML Score	Hybrid Score	Prediction
NTASWFTAL	0.33	0	Non-Allergen	0.56	0.56	Non-Toxin
LSPRWYFYY	0.29	0	Non-Allergen	0.55	0.55	Non-Toxin
SPRWYFYYL	0.29	0	Non-Allergen	0.55	0.55	Non-Toxin
NNAAIVLQL	0.29	0.5	Allergen	0.71	0.71	Toxin
DAALALLLL	0.41	0	Allergen	0.73	0.73	Toxin
LALLLLDRL	0.36	0	Non-Allergen	0.76	0.76	Toxin
KHWPQIAQF	0.4	0	Non-Allergen	0.62	0.62	Non-Toxin
LTYTGAIKL	0.32	0	Non-Allergen	0.7	0.7	Non-Toxin

## Data Availability

Data are contained within the article or Appendix A.

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
