# Peer review of "Immunoreactivity Analysis of MHC-I Epitopes Derived from the Nucleocapsid Protein of SARS-CoV-2 via Computation and Vaccination"

_vaccines, 2024, doi:10.3390/vaccines12111214_

Round 1

Reviewer 1 Report (Previous Reviewer 2)

Comments and Suggestions for Authors In my opinion, this version of the manuscript is better than the previous ones, however, there are still some points that need clarification and a few criticisms that should be addressed:   - Consistency with the abbreviations (e.g.: nucleocapsid protein N or NP)    - Line 344, for the reader it is hard to understand which sequences was used. The accession numbers of the analyzed sequences should be mentioned in the main text (the author mentioned the sequence n°1 in the supplementary table, but I didn't find it. In the supplementary table S1, there isn't a numerical order)   - Figure 5 and Paragraph 3.9: the authors should explicit the epitope used for the immunization, for example the readers don't know what is 9-mer_SARS2-NP_48 (if they don't read also the suplemmentary table 7). Moreover in the figure, the stimulation with 9-mer_SARS2-NP_48 is better than with 9-mer_SARS2-NP_104 and 9-mer_SARS2-NP_105, why did not the authors mention it?    - Clarify when and how much the peptides were used in the immunization protocol   - Check for punctuation and typos

Author Response

Reviewer 2 Report (Previous Reviewer 3)

Comments and Suggestions for Authors

The authors responded constructively to all my comments and made appropriate changes to their article. The article can be published as is.

Author Response

Thank you for your devotion to our submission.

Reviewer 3 Report (New Reviewer)

Comments and Suggestions for Authors

In the manuscript by  Jiang and colleagues, they used computer simulation to identify dominant MHC-I epitopes of the H-2 and HLA-I superfamilies. Using various computer programs, they identified  64 dominant epitopes for the H-2 haplotype and 238 dominant epitopes for the HLA-I haplotypes.  The authors identified 8 preferred epitopes based on immunogenicity and conservation, and were used in docking simulations with MHC-I alleles. Using ELISA, ELISpot assays, and flow cytometry, the authors showed that the preferred epitopes stimulated humoral and cellular immunity and enhanced cytokine secretion in mice.  While the science in the manuscript is very good, I believe it is incomplete.  There are also multiple typographical errors throughout the manuscript.  Additionally, some experiments lack proper controls.

Major comments:

1.     The major histocompatibility complex (MHC-I) molecules, which play a crucial role in antigen presentation to T cells, differ significantly between mice and humans, leading to different epitope recognition patterns.  However, studies have shown that certain epitopes from pathogens can be recognized by both mice and humans, especially when using transgenic mice with human MHC genes. As these studies were performed in mice, the authors can’t say whether these “preferred epitopes” are also dominant in humans.  Thus, experiments designed to analyze whether convalescent humans recognize these epitopes would greatly strengthen the findings of this manuscript.

2.     In the discussion, the authors discuss Chao and colleagues' results regarding immunodominant epitopes. This study used materials from convalescent SARS-CoV-2 patients; the authors indicated that the dominant peptide “KTFPPTEPK” was also picked up in the present study. However, I see nothing in that paper regarding this peptide. Is it possible that the authors were referring to the peptide “NTASWFTAL,” which was picked up in the present study?

3.     Lines 77-79: The sentence “Chao Hu et al. confirmed that N361-369 (KTFPPTEPK) has high immunogenicity and can induce a strong CD8+ T-cell response to SARS-CoV-2 [16]. Mauricio Menegatti Rigo et al. identified SPRWYFYYL as a conserved N epitope [17], whereas Janish Kumar et al. predicted five NP epitopes for vaccine construction [18].” can be combined to “Chao and colleagues confirmed that N361-369 (KTFPPTEPK) has high immunogenicity and can induce a strong CD8+ T-cell response to SARS-CoV-2 [16], Rigo and colleagues identified SPRWYFYYL as a conserved N epitope [17] and Kumar and colleagues predicted five NP epitopes for vaccine construction [18].

4.     The authors need to compare their findings better with those of previous studies. For example, Kumar's (reference 18) study identified five epitopes, but only one matches this study.

5.     Figure 5: In this Figure, the authors present the ELISPOT data. No negative controls are presented using unrelated peptides.

6.     Figure 7: In this Figure, the negative control used was PBS. This is not a proper negative control. The authors should have used mouse serum from a non-immunized mouse.  Please fix.

7.     There doesn’t seem to be any quantification of IL-2  and IFN-γ in the study.

Minor comments:

    1. Line 46: “11” should be written as “Eleven.”

    2.     Line 70: “Jigyasa Verma et al.” should be written as “Verma and colleagues

    3.     Line 72: Ibrahim Farhani et al. should be “Farhani and colleagues.” Please correct. 

    4.     Line 80-81: The authors state, "The NTASWFTAL epitope has been shown to have greater HLA affinity and broader coverage [19,20]."  My question here is, greater affinity than what?

    5.     Line 94: The sentence with, “NC_045512.2[28274.29533]) of SARS-CoV-2 was” should be modified with the strain name, “NC_045512.2[28274.29533]) of SARS-CoV-2 (Wuhan strain).”

    6.     Line 102: “11” should be changed to “Eleven.”

    7.     Line 119: “Betacoronavirus” should be changed to “β-coronavirus

    8.     Line 123: The sentence with “how they are preserved:” should be changed to “how conserved they are:

    9.     The sentence with “preferred epitopes" should be changed to "preferred epitopes."

    10. Line 150: “is” should be changed to “was.”

    11. Line 162: “affinity of SARS-CoV-2 is” should be changed to "affinity of the original SARS-CoV-2 strain is.”

    12. Line 208: “specific IL-2 monoclonal” should be changed to “specific anti-IL-2 monoclonal.

    13. Line 211: The following sentence appears to be missing something, “The spleen was ground and centrifuged.” Was the spleen ground through a mesh screen? If so, add it to the sentence.

    14. Line 212: Please define “split red liquid.”

    15. Line 217: Remove the period after “cultured.”

    16. Line 245: “experimental mice splenocytes” should be changed to “experimental mouse splenocytes.”

    17. Line 251: “solution. Incubate at 4oC…” should be changed to “solution. Cells were incubated at 4oC…”

    18. Line 255: “sequence. first gate…” should be changed to “sequence. The first gate….”

    19. Line 283: “of the novel” should be changed to “of the SARS-CoV-2

    20.  Line 284: In the sentence, “Table 3 shows the statistical results of all the dominant epitopes.” What do the numbers represent? This table requires a better explanation.

    21.  Line 286: Remove “and…

    22. Line 292-295: “Through detection, 177 of the 411 9-mer epitopes were found to be immunogenic. Among them, 47 epitopes have both immunogenicity and high affinity (Supplementary Table S5). Above all, we synthesized the affinity, immunogenicity, and conservation of all 9-mer epitopes of SARS-CoV-2 NP.”  should be changed to “Of the 411 9-mer epitopes, 177 were immunogenic. Among these 177 epitopes, 47 were both immunogenic and of high affinity (Supplementary Table S5). We synthesized the high affinity, immunogenic, and conserved 9-mer epitopes of SARS-CoV-2 NP.

    23. Line 380: In the sentence, “NNAAIVLQL, DAALALLLL not only toxic…” should be changed to “NNAAIVLQL, DAALALLLL were not only toxic….”

    24. Lines 402-404: In the sentence The SARS-CoV-2 NP-153 antigen epitope induced a good immune response in both types of mice, followed by SARS-CoV-2 NP-104 and SARS-CoV-2 NP-105.  My question is, why not include NP-48 with 2 NP-104 and SARS-CoV-2 NP-105?

    25. Lines 534-535: The sentence, “The homology between the variants in various regions and the Wuhan strain of SARS-CoV-2 (NC:045512) is also very low [53]” is somewhat deceiving.  In this paper by Ruiz and colleagues, they report a high level of identity between British and Spanish strains.  While there were mutations among strains, the identity was high overall. The Wuhan strain is not mentioned in this article. This sentence should be deleted. 

Comments on the Quality of English Language

I have provided corrections to some of the typographical errors.

Author Response

Response to Reviewer 3
In the manuscript by Jiang and colleagues, they used computer simulation to identify dominant
MHC-I epitopes of the H-2 and HLA-I superfamilies. Using various computer programs, they
identified 64 dominant epitopes for the H-2 haplotype and 238 dominant epitopes for the HLA-I
haplotypes. The authors identified 8 preferred epitopes based on immunogenicity and
conservation and were used in docking simulations with MHC-I alleles. Using ELISA, ELISpot
assays, and flow cytometry, the authors showed that the preferred epitopes stimulated
humoral and cellular immunity and enhanced cytokine secretion in mice. While the science in
the manuscript is very good, I believe it is incomplete. There are also
multiple typographical errors throughout the manuscript. Additionally, some experiments lack
proper controls.

I would like to express my heartfelt gratitude for your recognition and approval of our work.
Your feedback and suggestions have greatly improved the quality and clarity of the manuscript.
We have taken your comments into careful consideration and have made the necessary
revisions to improve the overall quality of our manuscript. Please refer to the revised
manuscript.

Comment 1: The major histocompatibility complex (MHC-I) molecules, which play a crucial
role in antigen presentation to T cells, differ significantly between mice and humans, leading to
different epitope recognition patterns. However, studies have shown that certain epitopes
from pathogens can be recognized by both mice and humans, especially when using
transgenic mice with human MHC genes. As these studies were performed in mice, the
authors can’t say whether these “preferred epitopes” are also dominant in humans. Thus,
experiments designed to analyze whether convalescent humans recognize these epitopes
would greatly strengthen the findings of this manuscript.

Response:
Thanks to referee’s comments and constructive proposal. In previous studies, we found cross
immunoreactivities of viral epitopes across genotypes, populations and species (PMID:
35455313). In this research, we observed a similar phenomenon through bidirectional
hierarchical clustering of affinity data, so that the reasonable speculation that preferred
epitopes are immunoreactive in the population is based on the same principle. Thanks to the
reviewers for their suggestions, we will also consider the trial of transgenic mice later. To be
more meticulous, we revised the conclusions to be explicitly limited to these epitopes in both
our validation and other articles. "The scope of application is limited, so specific epitope should
be further evaluated."

Comment 2: In the discussion, the authors discuss Chao and colleagues' results regarding
immunodominant epitopes. This study used materials from convalescent SARS-CoV-2
patients; the authors indicated that the dominant peptide “KTFPPTEPK” was also picked up in
the present study. However, I see nothing in that paper regarding this peptide. Is it possible
that the authors were referring to the peptide “NTASWFTAL,” which was picked up in the
present study?

Response: Thank you for your reminding. Our intention was indeed to write "NTASWFTAL", a
peptide found in the cited study, and we revised in the new version. " Chao and colleagues
confirmed that “NTASWFTAL” has high immunogenicity and can induce a strong CD8+ T-cell
response to SARS-CoV-2 [16].” “Among them, NTASWFTAL has been validated by Chao and
colleagues as a highly immunogenic peptide segment[16]”

Comment3: Lines 77-79: The sentence “Chao Hu et al. confirmed that N361-369
(KTFPPTEPK) has high immunogenicity and can induce a strong CD8+ T-cell response to
SARS-CoV-2 [16]. Mauricio Menegatti Rigo et al. identified SPRWYFYYL as a conserved N
epitope [17], whereas Janish Kumar et al. predicted five NP epitopes for vaccine construction
[18].” can be combined to “Chao and colleagues confirmed that N361-369 (KTFPPTEPK) has
high immunogenicity and can induce a strong CD8+ T-cell response to SARS-CoV-2 [16], Rigo
and colleagues identified SPRWYFYYL as a conserved N epitope [17] and Kumar and
colleagues predicted five NP epitopes for vaccine construction [18].

Response: Thank you for your valuable feedback. We have amended this paragraph as
followed: “Chao and colleagues confirmed that NTASWFTAL has high immunogenicity and can
induce a strong CD8+ T-cell response to SARS-CoV-2 [16]. Rigo and colleagues identified
SPRWYFYYL as a conserved N epitope [17] and Kumar and colleagues predicted five NP
epitopes for vaccine construction [18].”

Comment4: The authors need to compare their findings better with those of previous studies.
For example, Kumar's (reference 18) study identified five epitopes, but only one matches this
study.

Response: Thank you for your question. First of all, the latest version of the publicly
recognized algorithm is constantly being innovated and optimized. The Kumar's study used
the NetCTL (https://services.healthtech.dtu.dk/services/NetCTL-1.2/), which was updated in
2006 at the latest version. It has already been replaced by NetMHCpan series tools. As for our
work, 5 tools were enrolled, including the latest NetMHCpan. And we adopted the integration
principle in bioinformatics analysis (PMID: 36798136). This greatly improves the accuracy of
the analysis. Our validation focused on 8 “preferred epitopes” with high affinity, immunogenic,
and multi-MHC immunoreactivities. The results which match those of previous studies can be
regarded as cross-verified. However, this does not mean that those reported epitopes which
have not been verified by our research are necessarily not immunoreactive.

Comment5: Figure 5: In this Figure, the authors present the ELISPOT data. No negative
controls are presented using unrelated peptides.

Response: Thank you very much for your reminder. The values used in the analysis results
are the numbers of spots “minus” the numbers of spots of the negative control. This point has
been illustrated in the figure notes. “Figure 5. The values used in the analysis results are the
spots numbers of each pore minus the numbers of spots of the negative control. Validation of
dominant immunogenic epitopes from ELISpot experiments. Mouse spleen cells are
stimulated to mount cellular immune responses via the eight dominant epitopes. Blue
represents IL-2 secreted by BALB/c mice, and purple represents that secreted by C57BL/6
mice.” We used the S protein epitopes of SARS-CoV-2 and SARS as irrelevant peptide
controls. The results were almost indistinguishable from the blank controls we set up.
Therefore, we directly use the blank control results for calculation.

Comment6: Figure 7: In this Figure, the negative control used was PBS. This is not a proper
negative control. The authors should have used mouse serum from a non-immunized
mouse. Please fix.

Response: Thank you for pointing out the errors. We set up two types of negative controls in
the experiment (negative control for the experimental system and whether or not immunization
was performed, respectively), but only one of them is stated in the writing. We've rewritten this
section. “Overload of mouse serum was used as the positive controls. Mouse sera from a
non-immunized mouse or PBS with 1% BSA were used as the reaction background or system
negative control, respectively.”

Comment7: There doesn’t seem to be any quantification of IL-2 and IFN-γ in the study.

Response: This study was designed to explore immunoreactivity of the NP target peptide by
differential responses between the experimental and control groups. Hence, no absolute
quantitative experimental design was carried out. Considering referee’s suggestion, it is
constructive and would be more significant when evaluating the immunological application of
new vaccines or related peptides. We would enroll this part in future investigations.

Comment8: Line 46: “11” should be written as “Eleven.”

Response: Thanks for the suggestion, but I did not find this in the article line 46.

Comment9: Line 70: “Jigyasa Verma et al.” should be written as “Verma and colleagues”

Response: Thank you for pointing out the errors. We have modified the original text as you
requested. “Over the past few years, Verma and colleagues predicted three immunogenic NP
peptides with high population coverage, and good docking with HLA-I molecules was identified
[14].”

Comment10: Line 72: Ibrahim Farhani et al. should be “Farhani and colleagues.” Please
correct.

Response: Thank you for pointing out the errors. We have modified the original text as you
requested. “Farhani and colleagues used conserved SARS-CoV-2 NP epitopes for vaccine
production and obtained vaccines with high population coverage that induced good cellular
and humoral immunity.”

Comment11: Line 80-81: The authors state, "The NTASWFTAL epitope has been shown to
have greater HLA affinity and broader coverage [19,20]." My question here is, greater affinity
than what?

Response: Thank you for pointing out the errors. We have modified the original text. “The
NTASWFTAL epitope has been ascertained with HLA affinity and broad coverage [19,20].”

Comment12: Line 94: The sentence with, “NC_045512.2[28274.29533]) of SARS-CoV-2
was” should be modified with the strain name, “NC_045512.2[28274.29533]) of SARS-CoV-2
(Wuhan strain).”

Response: Thank you for pointing out the errors. We have modified the original text as you
requested. “As input for the sequential in silico analyses, the nucleocapsid protein (N,
accession number: NC_045512.2[28274.29533]) of SARS-CoV-2 (Wuhan strain)”

Comment13: Line 102: “11” should be changed to “Eleven.”

Response: Thank you for pointing out the errors. We have modified the original text as you
requested.

Comment14: Line 119: “Betacoronavirus” should be changed to “β-coronavirus”

Response: Thank you for your meticulous attitude, but NCBI Taxonomy used “Beta-”. We
have modified the original text as you requested. “The conservative evaluation standard
between species was β-coronavirus (taxi: 694002)”

Comment15: Line 123: The sentence with “how they are preserved:” should be changed to
“how conserved they are:”

Response: Thank you for pointing out the errors. We have modified the original text as you
requested. “The predicted epitopes can be classified into four categories on the basis of how
con-served they are”

Comment16: The sentence with “preferred epitopes" should be changed to "preferred
epitopes."

Response: Thank you for comment but I failed to understand what you meant.

Comment17: Line 150: “is” should be changed to “was.”

Response: Thank you for pointing out the errors. We have modified the original text as you
requested.

Comment18: Line 162: “affinity of SARS-CoV-2 is” should be changed to "affinity of the
original SARS-CoV-2 strain is.”

Response: Thank you for pointing out the errors. We have modified the original text as you
requested. “A positive RANK value indicates that the binding affinity of SARS-CoV-2 was
greater than that of the variant”

Comment19: Line 208: “specific IL-2 monoclonal” should be changed to “specific
anti-IL-2 monoclonal.”

Response: Thank you for pointing out the errors. We have modified the original text as you
requested. “which was coated with specific anti-IL-2 monoclonal capture antibodies (diluted to
5 µg/mL (1:200) with sterile PBS).”

Comment20: Line 211: The following sentence appears to be missing something, “The
spleen was ground and centrifuged.” Was the spleen ground through a mesh screen? If so,
add it to the sentence.

Response: Thank you for pointing out the errors. We have modified the original text as you
requested. “The spleen was ground through the mesh screen and centrifuged. After the red
blood cell lysis for 10 minutes”

Comment21: Line 212: Please define “split red liquid.

Response: Thank you for your question. It's a lack of professionalism in the words. We meant
to describe red blood cell lysis buffer We have modified the text as you requested. “The spleen
was ground by the mesh screen and centrifuged. After the red blood cell lysis for 10 minutes”

Comment22: Line 217: Remove the period after “cultured.”

Response: Thank you for pointing out the errors. We have modified the original text as you
requested.

Comment23: Line 245: “experimental mice splenocytes” should be changed to
“experimental mouse splenocytes.”

Response: Thank you for pointing out the errors. We have modified the original text as you
requested.

Comment24: Line 251: “solution. Incubate at 4oC…” should be changed to “solution. Cells
were incubated at 4oC…”

Response: Thank you for pointing out the errors. We have modified the original text as you
requested. “”

Comment25: Line 255: “sequence. first gate…” should be changed to “sequence. The first
gate….”

Response: Thank you for pointing out the errors. We have modified the original text as you
requested.

Comment26: Line 283: “of the novel” should be changed to “of the SARS-CoV-2”

Response: Thank you for pointing out the errors. We have modified the original text as you
requested.

Comment27: Line 284: In the sentence, “Table 3 shows the statistical results of all the
dominant epitopes.” What do the numbers represent? This table requires a better
explanation.

Response: Thank you for pointing out the errors. We have modified the original text as you
requested. Further interpretation of the meaning of Table 3. “Table 3 shows the number of
inter- or intra-conservation of predicted epitopes of SARS-CoV-2 NPs.”

Comment28: Line 286: Remove “and…”

Response: Thank you for pointing out the errors. We have modified the original text as you
requested.

Comment29: Line 292-295: “Through detection, 177 of the 411 9-mer epitopes
were found to be immunogenic. Among them, 47 epitopes have both immunogenicity
and high affinity (Supplementary Table S5). Above all, we synthesized the affinity,
immunogenicity, and conservation of all 9-mer epitopes of SARS-CoV-2 NP.” should be
changed to “Of the 411 9-mer epitopes, 177 were immunogenic. Among these 177
epitopes, 47 were both immunogenic and of high affinity (Supplementary Table S5). We
synthesized the high affinity, immunogenic, and conserved 9-mer epitopes of
SARS-CoV-2 NP.

Response: Thank you for pointing out the errors. We have modified the original text as you
requested. Sincerely appreciated to referee’s help on our language-editing.

Comment30: Line 380: In the sentence, “NNAAIVLQL, DAALALLLL not only
toxic…” should be changed to “NNAAIVLQL, DAALALLLL were not only toxic….”

Response: Thank you for pointing out the errors. We have modified the original text as you
requested.

Comment31: Lines 402-404: In the sentence “The SARS-CoV-2 NP-153 antigen epitope
induced a good immune response in both types of mice, followed by SARS-CoV-2 NP-104 and
SARS-CoV-2 NP-105. My question is, why not include NP-48 with 2 NP-104 and
SARS-CoV-2 NP-105?

Response: Thank you for pointing out the errors. The failure to include NP-48 in the
discussion was an oversight in our analysis of the results. We have modified the original text
as you requested. “The NNAAIVLQL antigen epitope induced a good immune response in both
types of mice, followed by NTASWFTAL, LSPRWYFYY and SPRWYFYYL.”

Comment32: Lines 534-535: The sentence, “The homology between the variants in
various regions and the Wuhan strain of SARS-CoV-2 (NC:045512) is also very low
[53]” is somewhat deceiving. In this paper by Ruiz and colleagues, they report a high level of
identity between British and Spanish strains. While there were mutations among strains, the
identity was high overall. The Wuhan strain is not mentioned in this article. This sentence
should be deleted.

Response: Thank you for pointing out the errors. We have modified the original text as you
requested, removed this paragraph, and changed the bibliographic order.

Thank you for your meticulousness and rigor. All the comments not only gave us advice on the
structure and soundness of the article, but also helped us correct mistakes in grammar and
writings. On behalf of the research crew, we would like to express our sincere appreciation and
respect.

Reviewer 4 Report (New Reviewer)

Comments and Suggestions for Authors

In this manuscript by Jian et al., authors performed an interaction analysis between SARS-CoV-2 nucleocapsid and antigen-presenting MHC I molecules from mouse and human to evaluate immunoreactivity potential of this viral protein. The manuscript writing is very poor. It suffers from poor description of the results (especially those obtained from bioinformatic analysis) and lack of reference to figures/tables in the main text. There is also an excessive use of scientific jargon and unspecific terms. This fact makes the manuscript very difficult to read, follow and understand. I provide some evidence below:

-       Supplementary Figure S1, labels from the “Y axis” are missing. The color-coded scale is missing a title/legend. In line 276 it is stated: “the intensity of the epitopes showed a regional distribution”. Do the authors refer to the Z score associated to the interaction of the epitope with each MHC haplotype? What a is a regional distribution? Does this mean that the affinity is epitope-dependent rather than haplotype-dependent? Please explain what is the meaning of this and be more specific.

-       Lines 280-289, authors explain that they are going to analyze the conservation of predicted epitopes in the first sentence. Then, in the next paragraph they refer to pan-MHC-I epitopes. Moreover, there is no explanation about what is the meaning of intraspecific or interspecific in this analysis. Table 3 has a caption that only consists in one sentence. Overall, there are zero intentions from the authors to explain what’s included in these tables.

-       Line 294, the sentence “we synthesized the affinity, immunogenicity, and conservation of all 9-mer epitopes” that makes no sense. Supplementary Figure S5 does not refer to any of these terms at some point. There are no indicated criteria about how 8 peptides were selected among the initial 47 peptides.

-       Line 305-314, here after indicating that 8 peptides are selected for analysis, authors suddenly move to analyze another set of 411 peptides. This is quite confusing. The manuscript need a better organization of the experiments.

-       Lines 316-333, authors list some peptides with good docking performance but there is no reference to any figure or table supporting this conclusion. The lack of reference to figures and tables happens frequently in multiple paragraphs, making the story very difficult to follow.

-       Lines 375-383, again, authors do not refer to any figure or table supporting these claims (which is Table 4). In general, all tables are poorly explained by the authors. They only use a short sentence saying an obvious description but there is no description of what is the meaning of column headings. For example, the term “MERCI” is used in Table 4 but it is not explained anywhere else what this parameter means, nor the text nor figure captions.

-       Lines 397-404, authors mention that mice were stimulated and secretion of IL-2 was measured after “two or three days”. The lack of a consistent time post-stimulation is concerning for this experiment. Moreover, ELISpot analysis shown in figure 7 is missing negative and positive controls.

-       Figure 6A indicates that “cytokines” are measured in the APC channel. Please be specific, and as explained in the caption, indicated that APC in Figure 6A correspond to IFN-gamma.

Overall, I recommend writing the manuscript from zero, choosing a logical workflow of experiments that can be easily followed across the manuscript. I suggest to be more specific and use homogeneous terms and definitions in the text, this will make it easier to read.

Other issues that I have identified:

-       Line 43-45, this sentence needs a reference.

-       Lines 542-543, this conclusion is not supported by the data included in this manuscript. Authors have determined that preferred epitopes can bind to antibodies against SARS-CoV-2 NP, but there is no proof that those antibodies have neutralizing activity to prevent infection.

-       Authors do not consider that escape mutations SARS-CoV-2 NP can also happen (reviewed in PMID: 38140214)

Comments on the Quality of English Language

English grammar is correct, but the writing style of this manuscript makes it very difficult to understand. 

Author Response

Response to Reviewer 4

In this manuscript by Jian et al., authors performed an interaction analysis between
SARS-CoV-2 nucleocapsid and antigen-presenting MHC I molecules from mouse and human
to evaluate immunoreactivity potential of this viral protein. The manuscript writing is very poor.
It suffers from poor description of the results (especially those obtained from bioinformatic
analysis) and lack of reference to figures/tables in the main text. There is also an excessive
use of scientific jargon and unspecific terms. This fact makes the manuscript very difficult to
read, follow and understand. I provide some evidence below:

I would like to express my heartfelt gratitude for your attention and approval of my
manuscript. Your feedback and suggestions have greatly improved the quality and clarity of my
work. We have taken your comments into careful consideration and have made the necessary
revisions to improve the overall quality of our manuscript. Please refer to the revised
manuscript.

Comment1: Supplementary Figure S1, labels from the “Y axis” are missing. The color-coded
scale is missing a title/legend. In line 276 it is stated: “the intensity of the epitopes showed a
regional distribution”. Do the authors refer to the Z score associated to the interaction of the
epitope with each MHC haplotype? What a is a regional distribution? Does this mean that the
affinity is epitope-dependent rather than haplotype-dependent? Please explain what is the
meaning of this and be more specific.

Response: Thank you for your comment. The Y-axis represents SARS-CoV-2 NP 9 peptide
epitopes, which are not aesthetically marked on the graph due to the large number of them.
From top to bottom, the 411 consecutive 9-mer peptides of the SARS-CoV-2 NP are shown.
We have added a title to the figure S1.To show them in their entirety would confuse the picture
display. We did use Z-Score to process the data. As for “regional distribution”, it means that
peptides divergently show their immune-reactivities to multiple haplotypes throughout the
full-length NP. It is just like what referee comprehend as a “epitope-dependent” manner. But
we did not intend to emphasize epitope-dependent rather than haplotype-dependent.

Comment2: Lines 280-289, authors explain that they are going to analyze the conservation
of predicted epitopes in the first sentence. Then, in the next paragraph they refer to pan-MHC-I
epitopes. Moreover, there is no explanation about what is the meaning of intraspecific or
interspecific in this analysis. Table 3 has a caption that only consists in one sentence. Overall,
there are zero intentions from the authors to explain what’s included in these tables.

Response: Thank you for your feedback and suggestion. We have included more interpretations of the content of the charts in this result to help readers better understand the content of this paper. “Table 3 shows the conservation of MHC-I-restricted dominant epitopes of SARS-CoV-2 NPs. Among them, there are two conserved peptide segments among multiple haplo-types. Eight dominant epitopes of SARS-CoV-2 NPs were Interspecies-Intraspecies+. None of them was Interspecies+Intraspecies-. According to the statistical analysis results of the multi-MHC-I epitopes, more peptides are interspecies-intraspecies conserved in human HLA molecules than in mouse H-2 molecules. The HLA-I restrictive advantage epitope is more conserved than the H-2 restrictive advantage epitope.” We illustrate interspecific and intraspecific conservation in the 2.3 method. “The evaluation standard for intraspecific protection is for SARS-related coronavirus (taxi: 694009), except for SARS-related acute respiratory syndrome coronavirus 2 (taxi: 2697049). The conservative evaluation standard between species was β-coronavirus (taxi: 694002), excluding SARS-related coronavirus (taxi: 694009). In the analysis, peptide sequences conserved between SARS-CoV-2 and humans (taxi: 9606) or mice (taxi: 10088) were also excluded, and the conserved E value was <10-5.” Conserved across multiple coronavirus strains. This makes it more relevant for subsequent application to immunological interventions or drug candidates.

Comment3: Line 294, the sentence “we the affinity, immunogenicity, and conservation of all
9-mer epitopes” that makes no sense. Supplementary Figure S5 does not refer to any of these
terms at some point. There are no indicated criteria about how 8 peptides were selected
among the initial 47 peptides.

Response: Thank you for your suggestion. Supplementary Table S5 illustrates how the 47
high-affinity, high-immunogenic epitopes were selected, rather than illustrating how the 8
preferred epitopes were selected. We selected the final 8 preferred epitopes after synthesizing
their coverage of MHC molecules worldwide. We replenished this section in the article and
Supplementary Table 6 accordingly. “We comprehend the high affinity, immunogenic, and
conserved 9-mer epitopes of SARS-CoV-2 NP and selected the final 8 preferred epitopes after
synthesizing their coverage of MHC molecules worldwide.”

Comment4: Line 305-314, here after indicating that 8 peptides are selected for analysis,
authors suddenly move to analyze another set of 411 peptides. This is quite confusing. The
manuscript need a better organization of the experiments.

Response: Thank you for your valuable feedback. Line 305-314 is a separate analysis of the
multiple properties of the epitopes, with the purpose of screening out the “preferred” dominant
epitopes with high potent. While the bi-directional hierarchical cluster reveals a comparative
analysis of NP peptides interacting to MHC-I molecules from a general aspects. It could be
regarded as a further exploration based on the above results.

Comment5: Lines 316-333, authors list some peptides with good docking performance but
there is no reference to any figure or table supporting this conclusion. The lack of reference to
figures and tables happens frequently in multiple paragraphs, making the story very difficult to
follow.

Response: Thank you for your reminder. We had Figure 3 in revised manuscript for molecular
docking. Furthermore, we added a new Supplementary Table 8 to show the average binding
energy analysis of eight preferred epitopes after docking with MHC molecules, so that the
reader can more intuitively understand the content of this paper. “Among the binding modes of
each epitope and different MHC molecules, we analyzed the top 10 most important binding
modes with the lowest binding energies (Supplementary Table S8).”

Comment6: Lines 375-383, again, authors do not refer to any figure or table supporting
these claims (which is Table 4). In general, all tables are poorly explained by the authors. They
only use a short sentence saying an obvious description but there is no description of what is
the meaning of column headings. For example, the term “MERCI” is used in Table 4 but it is
not explained anywhere else what this parameter means, nor the text nor figure captions.

Response: Thank you for your proposal. Our interpretation of Table 4 is indeed inadequate.
This has been revised in new version. In fact, MERCI is a predictive model of the tool itself
(please refer to https://webs.iiitd.edu.in/raghava/toxinpred2/algo.html), and we chose the
recommended model for prediction of epitope sensitization. Revised version refers to “Toxicity
predictions were performed via two models of the website (ML+Hybrid). Sensitivities were
predicted via the MERCI model for websites.” In addition to this we have also added and
modified notes for other tables.

Comment7: Lines 397-404, authors mention that mice were stimulated and secretion of IL-2
was measured after “two or three days”. The lack of a consistent time post-stimulation is
concerning for this experiment. Moreover, ELISpot analysis shown in figure 7 is missing
negative and positive controls.

Response: Thank you for raising this question. We have modified the original text as you
requested. “After two days, the secretion of IL-2 from the spleen cells was observed.” Other
reviewers have also mentioned the issue of control groups, and we have included a note about
negative controls in the figure notes. “The values used in the analysis results are the spots
numbers of each pore minus the numbers of spots of the negative control.” And positive
control develop too numerous spots to be analyzed. Therefore, it was not added to the figure.
The completed medium was used as the negative control. As a positive control, Con A (10
µg/ml) was used. We used the S protein epitopes SARS-Cov-2 as irrelevant peptide control.

Comment8: Figure 6A indicates that “cytokines” are measured in the APC channel. Please
be specific, and as explained in the caption, indicated that APC in Figure 6A correspond to
IFN-gamma.

Response: Thank you for raising this question. We've modified this sentence as you
requested “Expression of CD8+ T-cell cytokines IL-2, IFN-γ was examined using flow
cytometry(A)The gating graphs for CD8+ T cells (APC correspond to IFN-gamma.).”

Comment9: Line 43-45, this sentence needs a reference.

Response: Thank you for raising this question. Line 43-45, the citation for this sentence is
from the WHO statistical analysis and is consistent with the first citation, so we have adjusted
the order of the article. “Coronavirus disease 2019 (COVID-19) is an infectious disease
caused by severe acute respiratory syndrome coronavirus 2 (SARS-CoV-2). On 11 March
2020, the World Health Organization (WHO) declared this novel coronavirus pandemic Since
December 2019, more than 776 million cases and 7.06 million deaths have been recorded
globally, but the actual numbers are considered higher[1].”

Comment10: Lines 542-543, this conclusion is not supported by the data included in this
manuscript. Authors have determined that preferred epitopes can bind to antibodies against
SARS-CoV-2 NP, but there is no proof that those antibodies have neutralizing activity to
prevent infection.

Response: Thank you for raising this question. We have removed this conclusion from the
manuscript.

Comment11: Authors do not consider that escape mutations SARS-CoV-2 NP can also
happen (reviewed in PMID: 38140214)

Response: Thank you for raising this question. We have included the corresponding quote in
the original text. Although N proteins are highly conserved, progressive escape mutations in
the N sequence affect their conservation properties. This fact is a limitation for the
development of cross-reactive vaccines based on N proteins.(PMID: 38140214)

Round 2

Reviewer 3 Report (New Reviewer)

Comments and Suggestions for Authors

I have read the revised manuscript by Jiang and colleagues.  The authors have been very responsive to my comments. I believe that the manuscript is suitable for publication.

Comments on the Quality of English Language

The manuscript is much improved.

Author Response

Thanks for referee for the valuable feedback.

Reviewer 4 Report (New Reviewer)

Comments and Suggestions for Authors

After reading the new manuscript version, it has improved a little but most of the main issues remains. Here there are some examples.

- Authors explain that negative controls from ELISpot analysis are obtained using complete medium (lines 231-233). For this experiment, the correct negative control would be spleen cells from a mock (or blank) group. In Materials and Methods (lines 205-206), authors indicate that a group of mice is injected with the backbone vector. I do not understand why authors have excluded this group for ELISpot analysis.

- The manuscript still abuses of scientific jargon. For example, terms like "good immune response" are repeated several times in the manuscript. This should be avoided since there is no a universal convention about what a good immune response is. 

- Unspecific terms like "cytokines" in a cytometry dot plot are still being used. (Figure 6A, last dot plot). This was mentioned in my previous report and it has not been corrected.

This new manuscript version has been uploaded withing a few working days after my review. I suggest authors to take time to improve the manuscript considering my previous comments and avoid rushing. Manuscript should be resubmitted after a consirable amount of changes to the language style are made as the wording remains the same.

Comments on the Quality of English Language

Similarly to the previous versior, English grammar is correct but the wording is very poor. I advise to contact a professional English editing service to specifically review the use of scientific jargon and unspecific terms.

Author Response

Response to Reviewer 4

After reading the new manuscript version, it has improved a little but most of the main issues remains. Here there are some examples.

I would like to express my heartfelt gratitude for your attention and 2nd review of our submission. Your feedback and suggestions have greatly improved the quality and clarity of the manuscript. We have taken your comments into careful consideration and have made the necessary revisions to improve the overall quality of our manuscript. Please refer to the revised manuscript.

Comment1: Authors explain that negative controls from ELISpot analysis are obtained using complete medium (lines 231-233). For this experiment, the correct negative control would be spleen cells from a mock (or blank) group. In Materials and Methods (lines 205-206), authors indicate that a group of mice is injected with the backbone vector. I do not understand why authors have excluded this group for ELISpot analysis.

Response: Thank you for your comment. The control group settings in our previous version of ELlSpot were not clearly expressed. We have made changes to the original text. “The completed medium was used as the negative control in all four groups of splenocytes. As a positive control, Con A (10 µg/ml) was also used in all four groups of splenocytes. We used the S protein epitopes SARS-Cov-2 as an irrelevant peptide control.” We used the PBS-injected group of mice as a negative control instead of mice injected with backbone vector According to several previous studies, the use of PBS as a negative control with mice is injected with the backbone vector as a negative control does not produce a significant specific immune response.

Comment2:  The manuscript still abuses of scientific jargon. For example, terms like "good immune response" are repeated several times in the manuscript. This should be avoided since there is no a universal convention about what a good immune response is. 

Response: Thank you for your feedback and suggestions. We have consulted professional English editing to revise this paper thoroughly, in which the resubmission retained the re-editing history for your convenience to review. We hope that this round of revision will bring the paper up to your standard for publication. We have replaced all the scientific jargons, such as "good immune response", with professional academic descriptions throughout the manuscript. For instance, “cellular responses or antiviral protection” in section 2.9, “a stronger cellular response” in section 2.13, and “a strongest cytokine secretion” in section 3.9 and 3.10, please refer to the revised version.

Comment3: Unspecific terms like "cytokines" in a cytometry dot plot are still being used. (Figure 6A, last dot plot). This was mentioned in my previous report and it has not been corrected.

Response: Thank you for the reminder, we have made the necessary changes in the figure legend instead of the figure itself. “The expression of the CD8+ T-cell cytokines IL-2 and lFN-y was examined via flow cytometry, (A) Gating strategy diagram for CD8+ T cells sub-population and cytokine detection (the Cytokine-APC refers to a testimony of IFN-y gating).” We sincerely invite referee to notice that the panel A is regarded as a strategy plot. The APC channel gating plot stands for the detection protocol on both IFN-γ and IL-2, so it may be more suitable to use “Cytokine-APC” as a general description.

Comment4:  This new manuscript version has been uploaded withing a few working days after my review. I suggest authors to take time to improve the manuscript considering my previous comments and avoid rushing. Manuscript should be resubmitted after a consirable amount of changes to the language style are made as the wording remains the same.

Response: Thanks for your feedback. As required, we have thoroughly revised the manuscript including the issues the referee raised. We have invited a professional English editor to refine all the terminology to avoid jargons or inappropriate descriptions. We examined the article word-by-word from the beginning to the very end, and made up to 245 corrections to those mis-leading statements. Together with the editing assistances and approvals from the other three referees’, we hope that this round of revision will come to the standard of your recognition.

Thank you for your meticulousness and rigor. All the comments not only gave us advice on the structure and soundness of the article, but also helped us correct mistakes in writings. On behalf of the research crew, we would like to express our sincere appreciation and respect

This manuscript is a resubmission of an earlier submission. The following is a list of the peer review reports and author responses from that submission.

Round 1

Reviewer 1 Report

Comments and Suggestions for Authors

I'm really sorry, but the content of this article is much more fundamental than what I understood from reading the summary. Furthermore, it is not written in a simple and clear way. I am therefore not able to provide judgment on this article.

Comments on the Quality of English Language

The syntaxe is not correct, words are mising (sentence p 2, line 65 to 66), page 2 line 888 to 92, sentence is not complete, page 2, line 93: it is certainly necessary to put a comma instead of a period between the 2 sentences,

Author Response

Dear referee,

Thanks very much for taking your time to review this manuscript. We really appreciate all the comments raised by reviewers and editor. We have rechecked and revised the manuscript according to the suggestions of the editors and reviewers, and highlighted it as required. The “Conclusion” section has been added to the last paragraph of the “discussion”. And in our article, we added information about the version, server, etc. of the database used in the research. We have supplemented studies on the toxicity and sensitization of the preferred epitopes, and corrected all typos and misleading sentences in the manuscript. For a more intuitive understanding of this study, we have drawn graphic abstracts. Please refer to the revised manuscript.

Our point-by-point responses to the reviewers' comments are attached to this letter. The comments are reproduced, and our responses are given directly afterward. We would also like to thank editor for allowing us to resubmit a revised copy of the manuscript.

Response to Reviewer 1

I'm really sorry, but the content of this article is much more fundamental than what I understood from reading the summary. Furthermore, it is not written in a simple and clear way. I am therefore not able to provide judgment on this article.

Response:

We are sorry that the last version failed to lead the referee to quite understanding on our study. In briefs, it would not be regarded as a pure analysis based on bioinformatics methods. Herein, we would like to elaborate further on the innovation and importance in detail, so that referee could probably have a more straightforward understanding of this research. 

The WHO dashboard indicated that around 504,079,039 people were infected and

6,204,155 died from COVID-19 caused by different variants of SARS-CoV-2 (PMID37498146). However, the current research is more inclined to explore the humoral immune epitopes of virus-specific neutralizing antibodies, while ignoring the cellular immune killing effect induced by MHC-I epitopes (PMID38044868, 38953857). This will seriously lead to the lameness of SARS-CoV-2 antigen immunology research. Therefore, our study has timely and fully made up for the deficiency of immunological characteristics of the target antigen of SARS-CoV-2. It provides important guidance for the development of specific vaccines that are cross-reactive among ethics and regions and adaptable to a wide range of variants.

The "integration" principle has greatly advanced the research of computationl immunology, particularly the properties of pathogens (PMID37292203). By making full use of the principle, this study is not as fundamental as ordinary bioinformatics analysis. Five affinity analysis algorithms were integrated, which resulted in strong confidence in SARS-CoV-2 NP MHC-I immunoreactivities, claiming 64/238 H-2/HLA-I restricted epitopes. By exploring the pan-MHC-I reactiveness, principles in NP specific anti-viral cellular immunity were illustrated. On the other hand, the large number of SARA-CoV-2 variants makes the conservation analysis critical. In this study, conservative analysis and affinity analysis were organically integrated to explore the mutation coefficient of dominant epitope and even the influence of mutation on immune effect at the amino acid level. In summary, the focus of this study is not a simple epitope analysis, but an in-depth antigen immunoreactivity mining applicable to a wide range of mutant strains.

In the etiology research of SARS-CoV-2, the validation of the dominant target molecules is usually limited to the theoretical level, and has not been effectively verified at the cellular level or in vivo(PMID38974021, 37493168). In this study, vecor-immunized mouse spleen cells were used to detect the ability of the epitopes to activate immune responses at the individual cell level by co-incubation with the epitope molecules. This will directly and effectively verify the immune response effect of the dominant target molecules and the accuracy of bioinformatics research under the principle of "integration". 

It would be expected that referee could clearly and accurately grasp the innovation and significance of the study based on the above. And further communications are welcomed. At the same time, we have added the "Conclusion" section at the end of the manuscript, hoping to facilitate reading experience and better understanding. For a more intuitive understanding of this study, we have drawn graphic abstracts. Please refer to the revised manuscript. 

Comment1: The syntaxe is not correct, words are missing (sentence p 2, line 65 to 66), page 2 line 888 to 92, sentence is not complete, page 2, line 93: it is certainly necessary to put a comma instead of a period between the 2 sentences.

Response: Thank you for your reminder. We have fixed all the errors you listed. “The syntaxe is not correct, words are missing (sentence p 2, line 65 to 66).” We modified it to “Over the past few years, Jigyasa Verma et al. predicted three immunogenic NP peptides with high population coverage, and good docking with HLA-I molecules were identified.” “page 2 line 888 to 92, sentence is not complete.” We modified it to “To analyze where the amino acids vary between the SARS-CoV-2 strains and the differences in their affinity to the HLA molecules and the dominant 9-peptide, the protein sequences of reported isolated strains (84 nucleocapsid proteins in Supplementary Table S1) were obtained from NCBI GenBank.““page 2, line 93: it is certainly necessary to put a comma instead of a period between the 2 sentences.” We've replaced the punctuation. We have also re-updated our writing of the original text to give it a more professional look.

We would like to express our gratefulness to the editors and referees. We have carefully reviewed and made every effort to address all the comments. We hope that the revised version meets the requirements of publication for the esteemed journal.

Best Regards,

Authors

Reviewer 2 Report

Comments and Suggestions for Authors

In this manuscript, the authors investigated the immunoreactivity of MHCI and epitopes derived from nucleocapsid protein using in silico and in vivo/vitro approaches. 

The study is very interesting, but to help a better comprehension and improve the quality of the manuscript, the authors should consider these points:

- the authors should do a graphical summary or a scheme of their study. It is useful for the reader to understand the rationale. Moreover, in my opinion, the title should stress the use of different approaches.

- in the material and methods section several information lack. For example: *the authors should describe the version and the parameter setting used during their in silico analysis.

*The GenBank accession number of SARS-CoV-2 variants should be mentioned

*The vaccination protocol should be completely re-written: how many injections were performed (three?)? adjuvants? used antigens? Moreover, I do not understand how many mice was use in experiments and how are divided.

- Lines 102-105 are not clear. Please, explain better the criteria of your choice.

- Did the authors investigated also the possible toxicity and antigenicity of the selected epitopes? 

- Line 202 it is mentioned table 1. Is it correct?

- in paragraph 3.7 it is useful if the authors described the lineage of the variants.

- please, correct the typos. 

Author Response

Dear referee,

Thanks very much for taking your time to review this manuscript. We really appreciate all the comments raised by reviewers and editor. We have rechecked and revised the manuscript according to the suggestions of the editors and reviewers, and highlighted it as required. The “Conclusion” section has been added to the last paragraph of the “discussion”. And in our article, we added information about the version, server, etc. of the database used in the research. We have supplemented studies on the toxicity and sensitization of the preferred epitopes, and corrected all typos and misleading sentences in the manuscript. For a more intuitive understanding of this study, we have drawn graphic abstracts. Please refer to the revised manuscript.

Our point-by-point responses to the reviewers' comments are attached to this letter. The comments are reproduced, and our responses are given directly afterward. We would also like to thank editor for allowing us to resubmit a revised copy of the manuscript.

Response to Reviewer 2

In this manuscript, the authors investigated the immunoreactivity of MHCI and epitopes derived from nucleocapsid protein using in silico and in vivo/vitro approaches. 

The study is very interesting, but to help a better comprehension and improve the quality of the manuscript, the authors should consider these points:

I would like to express my heartfelt gratitude for your recognition and approval of our work.

Your feedback and suggestions have greatly improved the quality and clarity of the manuscript. We have taken your comments into careful consideration and have made the necessary revisions to improve the overall quality of our manuscript. For a more intuitive understanding of this study, we have drawn graphic abstracts. Please refer to the revised manuscript.  Response: First of all, thank you for approving our article. We have answered some of your suggestions for this article, and modified or deleted the content of the article!

Comment 1: The authors should do a graphical summary or a scheme of their study. It is useful for the reader to understand the rationale.

Response: Thank you for your suggestion. In our manuscript, we have supplemented a diagram of research pipeline. Please refer to the revised manuscript.

Figure 1. Flowchart of the experiment. - Visible in attached

Comment 2:  The title should stress the use of different approaches.

Response: Thank you for your advice on naming article titles. We've got a long title with a lot of content. Therefore, we call all computation methods “computation”. Our new title is called for “Immunoreactivity Analysis of MHC-I Epitopes Derived from the Nucleocapsid Protein of SARS-CoV-2 via Computation and Vaccination” Please refer to the revised manuscript. Comment3: In the material and methods section several information lack. For example: *the authors should describe the version and the parameter setting used during their in silico analysis.

Response: Thank you for your valuable feedback. Another reviewer asked the same question. We have shown the version of the database used for the prediction of epitopes and the parameter settings in the supplementary Table S3.

Comment4: The GenBank accession number of SARS-CoV-2 variants should be mentioned

Response: Thank you very much for your reminder. We have supplemented the GenBank accession number of SARS-CoV-2 variants in the supplementary Table S1. Please refer to the revised manuscript.

Comment5: The vaccination protocol should be completely re-written: how many injections were performed (three?)? adjuvants? used antigens? Moreover, I do not understand how many mice was use in experiments and how are divided.

Response: Thank you for pointing out the shortage. We have rewritten the vaccination protocol.  

“BALB/c and C57BL/6 mice at 8 weeks of age were obtained from the Laboratory Animal Centre of the Fourth Military Medical University. The mice were divided into four groups: the C57BL/6 experimental group, the BALB/c experimental group, and their respective blank control groups. Each experimental group contained six mice and each blank control group contained three mice. At weeks 0, 3, and 6, pVAX-NPSARS-CoV-2 plasmids were subcutaneously injected at a dose of 50 µ g per mouse. BALB/c and C57BL/6 mice injected with PBS only were used as controls. Tail vein blood was taken two weeks after each immunization, and the separated serum was used for subsequent immune experiments. After 3rd injection, we sacrificed two BALB/C mice and two C57BL/6 mice and their spleen cells were collected for the ELISpot experiments.”  

Comment6/7: Lines 102-105 are not clear. Please, explain better the criteria of your choice.

Response: Thank you for pointing out the errors. We've rewritten this section. 

“We selected peptides that scored in the top 2% of more than two databases(When the 9-peptide appears in only four databases), selected peptides that scored in the top 2% of more than three databases(When the 9-peptide appears in five databases).”

Comment8: Did the authors investigated also the possible toxicity and antigenicity of the selected epitopes?  

Response: Thank you for your attention. We have added studies of epitope toxicity to the article. The antigenicity of the epitope is mainly expressed in his binding to the corresponding receptor. The antigenicity of the epitope in binding to MHC class I molecules is mainly reflected in the immunogenicity and immunoreactivity of the epitope, which has been mentioned in our article. In addition to this, we also include the study of epitope sensitization, epitope-triggered autoimmune reactions, which are also a manifestation of epitope antigenicity. Therefore we did not make a separate study of antigenicity, but distributed it among our other studies.

2.8. Prediction of peptide toxicity and sensitization

Peptides have proven to be one of the most promising tools for the treatment and prevention of a wide range of diseases. However, its toxicity and sensitizing properties may lead to the development of a range of symptoms that can compromise the effectiveness of prevention and treatment. Therefore, studies of epitope toxicity and sensitization are important. We used network algorithms based on ToxinPred2 (ToxinPred2(iiitd.edu.i)) and AlgPred 2.0 (AlgPred2(iiitd.edu.in)) to test for toxicity and sensitization, respectively, of the screened “preferred epitopes”. Toxicity and lethality were set to standard thresholds (0.7 for negative toxicity; 0.4 for negative allergenicity). Machine learning models were used to output potential toxins and allergens. Epitopes were considered to be suspect if they exceeded a negative toxicity threshold of 0.7. Epitopes above a negative allergen threshold of 0.4 were considered to be suspected allergens and tabulated for statistical analysis.

3.8 Predicting dominant epitope toxicity and lethality

Toxicity and sensitization of 8 NP 9 peptides screened by bioinformatics network algorithm were analyzed separately. At the default threshold (0.7 for negative toxicity; 0.4 for negative sensitization), as can be seen from the results, LALLLLDRL were identified by the algorithm as being at risk of causing toxicity among the dominant epitopes screened previously. NNAAIVLQL, DAALALLLL not only toxic but also allergenic. All three peptides have low toxic effects, the sensitizing effect of DAALALLLL is also very low. Only NNAAIVLQL has a higher sensitization. Thus, despite the risks as sociated with its application, it is still an epitope with great potential.

Table 4. Comprehensive assessment table for toxicity and allergenicity of dominant epitopes

Pepitides

MERCI

Score

BLAST

Score

Hybrid

Score

Prediction

ML Score

MERCI

Score

BLAST

Score

Hybrid

Score

Prediction

NTASWFTAL

0.33

0

0

Non-Allergen

0.56

0

0

0.56

Non-Toxin

LSPRWYFYY

0.29

0

0

Non-Allergen

0.55

0

0

0.55

Non-Toxin

SPRWYFYYL

0.29

0

0

Non-Allergen

0.55

0

0

0.55

Non-Toxin

NNAAIVLQL

0.29

0.5

0

Allergen

0.71

0

0

0.71

Toxin

DAALALLLL

0.41

0

0

Allergen

0.73

0

0

0.73

Toxin

LALLLLDRL

0.36

0

0

Non-Allergen

0.76

0

0

0.76

Toxin

KHWPQIAQF

0.4

0

0

Non-Allergen

0.62

0

0

0.62

Non-Toxin

LTYTGAIKL

0.32

0

0

Non-Allergen

0.7

0

0

0.7

Non-Toxin

Comment9: Line 202 it is mentioned table 1. Is it correct?

Response: Thank you for your reminder. This was an error in our writing and we have removed this section. We sorry about this miswriting.

Comment10: In paragraph 3.7 it is useful if the authors described the lineage of the variants. Response: Thank you for pointing out the inadequacy of this section. We have placed an introduction to the genealogy of SARS-CoV-2 in the Supplementary Table S1. Our variants contain a total of four species in the genus β-coronavirus, the Severe acute respiratory syndrome coronavirus 2, Bat coronavirus, Sarbecovirus sp. and Severe acute respiratory syndrome-related coronavirus. Their numbers are 8, 5, 70 and 1. This is something we have changed accordingly in the article 

Comment11: Please, correct the typos.

Response: Thank you for pointing out the errors. We have re-checked the entire article to correct or remove spelling errors, incorrect application of determiners and non-specialist vocabulary, which you can see in our re-submitted article!

We would like to express our gratefulness to the editors and referees. We have carefully reviewed and made every effort to address all the comments. We hope that the revised version meets the requirements of publication for the esteemed journal.

Best Regards,

Authors

Reviewer 3 Report

Comments and Suggestions for Authors

Since the emergence of SARS-CoV-2, the focus has been not only on neutralizing antibodies, but also on CD8 T cells that specifically bind to target cells with the involvement of MHC I proteins.  Therefore, understanding the immunogenicity of SARS-CoV-2 NP-specific MHC I-restricted epitopes is of paramount importance. Currently, at the initial stage of these studies, in silico bioinformatics methods are widely used to predict epitopes, which determine the direction of various in vitro/in situ immunological experiments and subsequent vaccine development. Thus, the topic of the article is relevant and its results may be useful to the readers of Vaccines. In general, I have a positive assessment of the methods and results of this study, but I have some unprincipled comments:

1) Introduction. Clearly state the purpose of the study at the end of this section.

(2) Materials and Methods. In the first subsection, it is desirable to briefly describe the characteristics of the high-performance computing equipment used by the authors (including overall performance), and to list the programs, servers, and databases used.

(3) Subsection 2.9. C57 is not an inbred mouse line, but rather its family, which includes C57Bl/6, C57BL/10, and others. Therefore, please indicate which specific line of the C57 family you have used in your experiments.

(4) Subsection 3.1. "These are described in the Methods section (Table 1, 2)." The meaning of this sentence is unclear, since Tables 1 and 2 refer to Section 3. "Results".

(5) Figure 4. "Balbc" needs to be corrected to Balb/c.

(6) Discussion. At the end of the study, it is desirable to indicate that the final product, directly suitable for vaccine development and other uses, can only be obtained in subsequent in vitro/in situ studies.

(7) The Discussion section is very long and not clearly focused on the main results of the study, therefore it is advisable to include an additional section "Conclusions".

(8) References need to be adapted to the MDPI style.

Author Response

Dear referee,

Thanks very much for taking your time to review this manuscript. We really appreciate all the comments raised by reviewers and editor. We have rechecked and revised the manuscript according to the suggestions of the editors and reviewers, and highlighted it as required. The “Conclusion” section has been added to the last paragraph of the “discussion”. And in our article, we added information about the version, server, etc. of the database used in the research. We have supplemented studies on the toxicity and sensitization of the preferred epitopes, and corrected all typos and misleading sentences in the manuscript. For a more intuitive understanding of this study, we have drawn graphic abstracts. Please refer to the revised manuscript.

Our point-by-point responses to the reviewers' comments are attached to this letter. The comments are reproduced, and our responses are given directly afterward. We would also like to thank editor for allowing us to resubmit a revised copy of the manuscript.

Response to Reviewer 3

Since the emergence of SARS-CoV-2, the focus has been not only on neutralizing antibodies, but also on CD8 T cells that specifically bind to target cells with the involvement of MHC I proteins.  Therefore, understanding the immunogenicity of SARS-CoV-2 NP-specific MHC I-restricted epitopes is of paramount importance. Currently, at the initial stage of these studies, in silico bioinformatics methods are widely used to predict epitopes, which determine the direction of various in vitro/in situ immunological experiments and subsequent vaccine development. Thus, the topic of the article is relevant and its results may be useful to the readers of Vaccines. In general, I have a positive assessment of the methods and results of this study, but I have some unprincipled comments:

I would like to express my heartfelt gratitude for your attention and approval of my manuscript. Your feedback and suggestions have greatly improved the quality and clarity of my work. We have taken your comments into careful consideration and have made the necessary revisions to improve the overall quality of our manuscript. For a more intuitive understanding of this study, we have drawn graphic abstracts. Please refer to the revised manuscript.  Response: First of all, we would like to thank you for liking our article, and at the same time, we have responded to some of your suggestions for this article, and modified or deleted the content of the article!

Comment1: Introduction. Clearly state the purpose of the study at the end of this section. Response: Thank you for your suggestion. We have stated the purpose of the study at the end of this section. “Research on the immunological properties of SARS-CoV-2 NP is being conducted while laying the foundation for the development of future SARS-CoV-2

immunization and related vaccines against SARS-CoV-2”.

Comment2: Materials and Methods. In the first subsection, it is desirable to briefly describe the characteristics of the high-performance computing equipment used by the authors (including overall performance), and to list the programs, servers, and databases used.

Response: Thank you for your feedback and suggestion. We have listed the programs, servers, and databases used in the Supplementary Table S3. Our computing is based on a server system that runs on the database site itself, and we don't need to assemble a high-performance computing equipment.

Comment3: Subsection 2.9. C57 is not an inbred mouse line, but rather its family, which includes C57Bl/6, C57BL/10, and others. Therefore, please indicate which specific line of the C57 family you have used in your experiments.

Response: Thank you for your suggestion. We have changed all sections of the article showing C57 to C57BL/6.

Comment4:  Subsection 3.1. "These are described in the Methods section (Table 1, 2)." The meaning of this sentence is unclear, since Tables 1 and 2 refer to Section 3. "Results". Response: Thank you for your valuable feedback. This was an error in our writing and we have removed this section.

Comment5:  Figure 4. "Balbc" needs to be corrected to Balb/c.

Response: Thank you for your reminder. We've already solved that problem.

Figure visible in attached

Comment6: Discussion. At the end of the study, it is desirable to indicate that the final product, directly suitable for vaccine development and other uses, can only be obtained in subsequent in vitro/in situ studies.

Response: Thank you for your proposal. We have indicated that the final product, directly suitable for vaccine development and other uses, can only be obtained in subsequent in vitro/in situ studies at the end of our study.

Comment7: The Discussion section is very long and not clearly focused on the main results of the study; therefore it is advisable to include an additional section "Conclusions".

Response: Thank you for raising this question. We have added an additional section “Conclusions” at the end of our study.

Considering the important role of nucleocapsid (NP) in the immunology of SARS-CoV-2 and the value of its clinical application, we performed bioinformatics analysis using various databases to screen 9-mer peptides with high affinity and high immunogenicity. Then the inter-&intra-species conservatism analyses screened and obtained 8 preferred epitopes. Hierarchical clustering analysis shown the similarities in interactions between different MHC molecules, superfamilies, and even across species. Molecular docking allowed visualization of the docking of MHC molecules to epitopes. In addition with the experimental validation and cross-lineage responsiveness exploration, antigenic properties conveyed the anti-viral applicability of NP in SARS coronavirus prevention and control.

Comment8:  References need to be adapted to the MDPI style.

Response: Thank you for raising this question. We have changed the references to MDPI style.

We would like to express our gratefulness to the editors and referees. We have carefully reviewed and made every effort to address all the comments. We hope that the revised version meets the requirements of publication for the esteemed journal.

Best Regards,

Authors

Round 2

Reviewer 2 Report

Comments and Suggestions for Authors

The authors replied exhaustively to all my requests.

I have only minor concerns:

- in my opinion, the graphical abstract is very confusing. Is it possible to simplify?

- In materials and methods section, lack the description of the production of the antigen used during in vivo experimentation

- line 325-328, please cite Table S1 and check the numbers (it is better use directly the GenBank accession number)

- in my opinion, table S7 should be in the main text and tables 1and 2  in the supplementary

- line 478, please add the reference

- there are again several typos and grammatical errors

- please, improve the resolution of the figures

Author Response

Response to the reviewer 2’s comments:

I would like to express my heartfelt gratitude for your recognition and approval of our work. Your feedback and suggestions have greatly improved the quality and clarity of the manuscript. We have taken your comments into careful consideration and have made the necessary revisions to improve the overall quality of our manuscript. Please refer to the revised manuscript.

The authors replied exhaustively to all my requests. I have only minor concerns:

Response: First of all, thank you for recognizing our research; we have resolved all of the suggestions for article. Please refer to the revised manuscript.

Comment 1:  in my opinion, the graphical abstract is very confusing. Is it possible to simplify?

Response: Thank you for your suggestion. After our in-depth thinking, we also agree with the reviewer's opinion. However, due to time, we could not design a perfect graphic abstract in a short time. Therefore, we remove this part of the content.

Comment 2:  In materials and methods section, lack the description of the production of the antigen used during in vivo experimentation.

Response: Thank you for your advice. We have reincorporated a description of the antigen production used during the in vivo experiments: “The pVAX-NPSARS-CoV-2 vector was constructed in our laboratory. The gene encoding the SARS-CoV-2 NP was subjected to gene synthesis by TSINGKE (Tsingke Biotech Co., Ltd., Beijing) based on a designed sequence. BamH I cleavage site was introduced upstream and Xho I cleavage site was introduced downstream of the sequence. The SARS-CoV-2 NP gene was inserted into a pVAX1 vector to construct the pVAX-NPSARS-CoV-2 vector. The absence of mutations was verified by sequencing. After sequencing, it was confirmed that there was no mutation. The plasmid was purified using the Plasmid Maxi kit (DP117, Chinese TIANGEN) and stored at −20℃ until use. Eight H2 restricted immunogenic dominant epitopes of SARS-CoV-2 NP were artificially synthesized (ChinaPeptides, China)”.

Comment3:  line 325-328, please cite Table S1 and check the numbers (it is better use directly the GenBank accession number)

Response: Thank you for valuable feedback. As you requested, we have revised this paragraph: “Our variants contain a total of four species in the genus β-coronaviruses, the severe acute respiratory syndrome coronaviruses 2, bat coronaviruses, sarbecoviruses sp, and Severe acute respiratory, syndrome-related coronavirus. The number of variants they contain are 8, 5, 70 and 1(Supplementary Table S1.)”. We applied strain names to describe variants in this paragraph instead of Accession number, because each of the 84 variants has its own unique Accession number, and it would take too much space to describe all of them. We put their Accession numbers in the supplementary material. Please find the information of variants in the Table S1.

Comment4: in my opinion, table S7 should be in the main text and tables 1and 2 in the supplementary.

Response: Thank you very much for your suggestion. Table S7 lists all the preferred epitopes. At the same time, some of the above epitope sequences appear in the multiple sequence alignment in Figure 3A. Therefore, we did not have all the preferred epitope sequences recurring in the main text, but in the form of supplementary material. On the other hand, Tables 1 and 2 represent the results of multi-database & genotype affinity analyses, which is the key process of epitope screening. In order to make the article more logical and complete, we think it is more reasonable for tables 1 and 2 to appear in the main text.

Comment5:  line 478, please add the reference

Response: Thank you for pointing out the shortage. We have added the reference. “SARS-CoV-2 originates from bats and other mammals (Santos-López, G.; Cortés-Hernández, P.; Vallejo-Ruiz, V.; Reyes-Leyva, J. SARS-CoV-2: basic concepts, origin and treatment advances. Gaceta medica de Mexico 2021, 157, 84-89, doi:10.24875/gmm.M21000524.)”

Comment6:  there are again several typos and grammatical errors.

Response: Thank you for pointing out the errors. Once again, we have corrected the typos. Please refer to the revised manuscript.

Comment7: please, improve the resolution of the figures.

Response: Thank you for your attention. We have improved the resolution of the figures.
